# FLOW DISTORTED PLANE WAVES

## ABSTRACT

The plane wave basis is widely used in Galerkin approximation, due to its periodicity and computational advantage, where the fast Fourier transform (FFT) can be applied. However, since its spatial resolution is uniform, the number of basis functions required can be excessive for problems with rapidly varying local features. We propose an adaptive basis called flow-distorted plane wave (FDPW), where the bijection of a normalizing flow is used to distort the problem domain, hence achieving adaptive resolution. We apply FDPW to Kohn-Sham density functional theory (DFT) calculations to both molecular and solid-state systems, demonstrating improved speed and memory usage. [1]

## 1 INTRODUCTION

Kohn–Sham density functional theory (DFT) Hohenberg & Kohn (1964); Kohn & Sham (1965) is the workhorse for electronic structure in molecules and solids due to its optimal trade off between efficiency and approximation power. Plane waves (PW) are attractive for their simplicity and FFT-based efficiency, but their uniform spatial resolution can require many basis functions near nuclei or other localized features; localized orbitals alleviate this but complicate periodic calculations.

We introduce flow-distorted plane waves (FDPW): a Galerkin basis obtained by composing a bijective, periodic normalizing flow on the 3-torus with the usual PW coordinates. The resulting map adapts resolution to the electronic structure while retaining PW algebra. A modified Bloch phase $e^{i\mathbf{k}^\top f^{-1}(\mathbf{r})}$ preserves k-point orthogonality and decoupling in periodic systems, and type-2 NUFFT (nonuniform Fast Fourier Transform) enables fast transforms on the distorted grid. FDPW applies to both non-periodic and periodic systems within ab-initio Kohn–Sham DFT. Our **contributions** are: (a) Parameter-efficient adaptive PW: represent the distortion with a small neural flow instead of PW coefficients as in prior distorted PWs (DPW) Gygi (1993), greatly reducing parameters while maintaining spectral accuracy; (b) Unified periodic/non-periodic formulation: extend DPW to arbitrary lattices via an affine cell map and a Bloch phase that maintains k-space orthogonality and decoupling; (c) Practical gains: with prescribed-density initialization and an NUFFT-based implementation, FDPW reaches target accuracy with fewer basis functions than standard PWs on molecules and crystalline solids, yielding speed and memory improvements.

## 2 RELATED WORKS

### 2.1 ADAPTIVE BASIS SET IN AB-INITIO ELECTRONIC STRUCTURE MODELING

The formulation of distorted plane waves was introduced by François Gygi in Gygi (1993). The calculation was done on the DPW basis, so mathematically it is quite similar to the regular PW calculation in reciprocal space. Subsequently, it was applied to molecular dynamics (MD) Gygi & Galli (1995). Later on, real-space formulations are proposed Gygi (1995); Zumbach & Maschke (1983); Zumbach et al. (1996); Modine et al. (1997). Also related is the local-scaling method Bokanowski & Grébert (1996). Recently, Lindsey and collaborators proposed a spectrally accurate, "diagonal" adaptive basis for periodic systems Lindsey & Sharma (2024).

On the other hand, there have been efforts to use normalizing flow for orbital free DFT de Camargo et al. (2023) and solid-state calculation Wirnsberger et al. (2022).

### 2.2 NEURAL NETWORK ANSATZ IN QUANTUM SIMULATION

Neural VMC represents antisymmetric many-electron wavefunctions with expressive networks and optimizes them by stochastic energy minimization. Examples include FermiNet and transformer/DP

---

[1]Our code will be open-sourced later.

variants Pfau et al. (2020); von Glehn et al. (2023); Pham et al. (2023); Li et al. (2022); Gerard et al. (2022). Neural quantum states further extend to lattice/continuous and periodic settings Vivas et al. (2022); Zhao et al. (2023); Yoshioka et al. (2021); Pescia et al. (2022); Luo & Halverson (2023).

## 3 PRELIMINARIES

### 3.1 SOLVING DFT WITH GALERKIN APPROXIMATION

Kohn-Sham DFT solves the following eigenvalue problem called the Kohn-Sham equation $\hat{H}[\rho]\,|\psi_n\rangle = \varepsilon_n\,|\psi_n\rangle$ where the Hamiltonian matrix is given by $H_{nm}[\rho] = \langle\psi_n|\hat{H}[\rho]|\psi_m\rangle$ and the eigenstates $\{\psi_n\}$ are the ground-state orbitals. Under Galerkin approximation, the orbital $\psi_n$ are represented as the basis coefficients $\mathbf{c}_n \in \mathbb{C}^{N_{\text{basis}}}$ where $c_{np} = \langle\phi_p|\psi_n\rangle$, and the Hamiltonian operator is represented with the matrix element $H_{pq}[\rho] = \langle\phi_p|\hat{H}[\rho]|\phi_q\rangle$, the infinite-dimensional eigenvalue problem is converted to the following finite-dimensional eigenvalue problem (see Appendix A.1)

$$H_{nm}[\rho] = \sum_{pq} c_{np}^* c_{mq} H_{pq}[\rho], \tag{1}$$

which can be solved via either via Self-Consistent-Field (SCF) iteration or the direct minimization of the Rayleigh quotient. We will use atomic unit through out this paper, where the length unit is Bohr and the energy unit is Hartree (Ha) unless otherwise stated.

### 3.2 PERIODIC SYSTEM AND FFT

Solid-state physics deals with periodic structures that can be described by a Bravais lattice (Appendix A.2). Different from the non-periodic/finite system, the orbital index becomes a composite $(n, \mathbf{k})$ where $n$ is the band index, and $\mathbf{k}$ is a point in the Brillouin zone (BZ). Furthermore, the form of the orbitals $|\psi_{n\mathbf{k}}\rangle$ are dictated by the Bloch's theorem Bloch (1929) $\psi_{n\mathbf{k}}(\mathbf{r}) = \exp[i\mathbf{k}^\top\mathbf{r}]u_{n\mathbf{k}}(\mathbf{r})$. where $u_{n\mathbf{k}}(\mathbf{r})$ is a function periodic over the unit cell. When applying Galerkin approximation, one expand the periodic part $u_{n\mathbf{k}}$ with a basis $|\phi_p\rangle$ periodic on the unit cell $\Omega_A$, and the basis coefficients is $\mathbf{c}_{n\mathbf{k}} \in \mathbb{C}^{N_{\text{basis}}}$ where $c_{n\mathbf{k}p} = \langle\phi_p|u_{n\mathbf{k}}\rangle$, $N_{\text{band}}$ is the number of bands and $N_k$ is the number of k-points. The density is determined by the periodic part $u_{n\mathbf{k}}$ only: $\rho(\mathbf{r}) = \sum_{n\mathbf{k}} f_{n\mathbf{k}}|\psi_{n\mathbf{k}}(\mathbf{r})|^2 = \sum_{n\mathbf{k}} f_{n\mathbf{k}}|u_{n\mathbf{k}}(\mathbf{r})|^2$. The KS eigenvalues $\varepsilon_{n\mathbf{k}}$ are also referred to as the band structure. For more details, see Appendix A.3.

In PW basis, $c_{n\mathbf{k}\mathbf{G}} = \langle e^{i\mathbf{G}^\top\mathbf{r}}|u_{n\mathbf{k}}\rangle$ is the Fourier transform of $u_{n\mathbf{k}}(\mathbf{r})$, therefore we can evaluate on uniform $\mathbf{r}$-space grid $\{\mathbf{r}_i\}_{i=1}^N$ (see Appendix D), via FFT: $\mathbf{u}_{n\mathbf{k}} = \frac{N}{\sqrt{\Omega_A}}\text{FFT}^{-1}(\mathbf{c}_{n\mathbf{k}}) \in \mathbb{C}^{N_{\text{basis}}}$ where $u_{n\mathbf{k}i} = u_{n\mathbf{k}}(\mathbf{r}_i)$ and $N$ is the total FFT grid size, which arises since we use the default `numpy` FFT normalization convention which multiplies $1/N$ for inverse FFT. The reciprocal vectors $\mathbf{G_n} = \mathbf{B}(n_1 \quad n_2 \quad n_3)^\top$ are reciprocal lattice points where $\mathbf{n}$ is the lattice indices, and when we write $\mathbf{G}$ in subscript we mean indexing by the reciprocal lattice indices $\mathbf{n} = (n_1, n_2, n_3)$. Similarly $\mathbf{k_m} = \mathbf{B}\left(\frac{m_1}{M_1} \quad \frac{m_2}{M_2} \quad \frac{m_3}{M_3}\right)$ are k-points in BZ which is on the direct lattice, and subscript $\mathbf{k}$ means indexing via the direct lattice index $\mathbf{m} = (m_1, m_2, m_3)$.

### 3.3 DIFFERENTIABLE GEOMETRY

Differential geometry (DG) describes calculus on a smooth manifold. Smooth bijections $g : \Xi \to X$ are diffeomorphisms between manifolds. Pullbacks $T^*g, T^*g^{-1}$ define ways to map covariant objects between manifolds (e.g. densities, scalar fields), while pushforwards $Tg, Tg^{-1}$ map contravariant objects (e.g. $v^\alpha\partial_\alpha$) between manifolds. The pullbacks and the pushforward usually carry a Jacobian factor, except for 0-forms/scalar fields (e.g., potentials), whose pullback is simply function composition: $T^*f^{-1}(\phi) = \phi \circ f^{-1}$. For operators, we consider their action on objects. For example, the pullback of the Laplacian $\Delta\phi = \boldsymbol{\nabla}\cdot\boldsymbol{\nabla}\phi$ acting on scalar fields $\phi$ under $T^*g^{-1}$ is the Laplace-Beltrami operator $|J|^{-1}\partial_\alpha\left(|J|g^{\alpha\beta}\partial_\beta\phi\right)$ (see Appendix E.6). For a quick recap on DG, see Appendix E. We will use Einstein notation throughout this paper.

### 3.4 NORMALIZING FLOW ON CIRCLES $\mathbb{S}^1$

Normalizing flows is a technique for defining a complex distribution $p$ from a simple distribution $p_0$ by distorting its probability density via a *bijection* $g : \Xi \to X$:

$$p(\mathbf{x}) = |J|^{-1} p_0(g^{-1}(\mathbf{x})), \quad \boldsymbol{\xi} = g^{-1}(\mathbf{x}) \sim p_0. \tag{2}$$

where $J = \frac{\partial x^i}{\partial \xi^\alpha}$ is the Jacobian. To avoid ambiguity, throughout this paper the indices for $\xi$ are in Greek letters and indices for $x$ are in Roman letters. In the language of DG, $g$ is a diffeomorphism and the above change of variable formula arises from the pullback on the density bundle $T^* f^{-1}(\mathrm{d}^n \xi) = |J|^{-1} \mathrm{d}^n x$, which ensures invariance of $L^1$ norm.

As shown in Rezende et al. (2020), by fixing the last knot $(x^{(K)}, y^{(K)})$ and gradient $\delta^{(K)}$ to be the same as the first knot $(x^{(0)}, y^{(0)})$ and gradient $\delta^{(0)}$, the Rational-quadratic spline (RQS) bijection (see Appendix C) used in neural spline flow Durkan et al. (2019) becomes a bijection on the circle $\mathbb{S}^1$, which can then be used to construct bijections on the $D$-dimensional torus $\mathbb{T}^D \cong (\mathbb{S}^1)^D$. We refer to this modified RQS as the circular RQS.

### 3.5 Distorted plane wave

Distorted plane waves (DPW) were first proposed in Gygi (1993). Take a 3-torus $\Omega = [0, a]^3$ as the unit cell, where $a$ is the size of the fundamental domain. Given a *bijection* $g : \Omega \to \Omega$ on that satisfies periodic boundary condition, DPW is the pullback of the plane wave in the parameter space $\langle \boldsymbol{\xi} | \mathbf{G} \rangle := \phi_{\mathbf{G}}(\boldsymbol{\xi}) = \frac{1}{\sqrt{\Omega}} \exp\left[i\mathbf{G}^\top \boldsymbol{\xi}\right]$ to the physical space $\mathbf{x}$:

$$\langle \mathbf{x} | \mathbf{G} \rangle := \phi_{\mathbf{G}}(\mathbf{x}) = \frac{1}{\sqrt{\Omega}} |J|^{-\frac{1}{2}} \exp\left[i\mathbf{G}^\top g^{-1}(\mathbf{x})\right]. \tag{3}$$

The factor $|J|^{-\frac{1}{2}}$ arises naturally from the pullback on the *half-density bundle* $T^* f^{-1}(|\mathrm{d}^n \xi|^{\frac{1}{2}}) = |J|^{-\frac{1}{2}} |\mathrm{d}^n x|^{\frac{1}{2}}$, which ensures invariance of $L^2$ norm (see Appendix E.9). Half-densities are $L^2$ normalized functions like the wavefunctions and their basis, which are $(0, \frac{1}{2})$-tensors. Furthermore, the pullback on the half-density bundle is unitary, which means the orthonormality of $\boldsymbol{\xi}$-space PW still holds in $\mathbf{x}$-space after the pullback (see proof at Appendix B), and we can write $\langle \mathbf{G} | \mathbf{G}' \rangle = \delta_{\mathbf{G}\mathbf{G}'}$ without ambiguity since the orthonormality does not depend on the coordinate system.

From section 3.2, under PW basis, the periodic part of the Bloch wave $u_{n\mathbf{k}}(\mathbf{r})$ can be evaluated on a uniform $\mathbf{r}$-space grid via FFT. Under the DPW basis, given a uniform $\boldsymbol{\xi}$-space grid $\{\boldsymbol{\xi}_i\}_{i=1}^N$, we have a distorted $\mathbf{x}$-space grid $\{g(\boldsymbol{\xi}_i)\}_{i=1}^N$. Similarly, given DPW coefficients $c_{n\mathbf{k}\mathbf{G}} = \langle \phi_{\mathbf{G}} | u_{n\mathbf{k}} \rangle$, we can evaluate $u_{n\mathbf{k}}(\mathbf{x})$ on the distorted grid via FFT: $\mathbf{u}_{n\mathbf{k}} = \frac{N}{\sqrt{\Omega}} \mathbf{J}^{-\frac{1}{2}} \mathrm{FFT}^{-1}(\mathbf{c}_{n\mathbf{k}}) \in \mathbb{C}^{N_{\text{basis}}}$ where $u_{n\mathbf{k}i} = u_{n\mathbf{k}}(g(\boldsymbol{\xi}_i))$ and $J_i^{-\frac{1}{2}} = |J(\boldsymbol{\xi}_i)|^{-\frac{1}{2}}$. Note that the density of DPW is generated both from the distortion $g$, and the unitary transformation of basis via $c_{n\mathbf{k}\mathbf{G}}$. The distortion effectively creates a non-uniform spatial resolution, which can reduce the required basis set cutoff, since usually high-frequency components are localized in the ground-state solution of solid-state systems. This brings performance gain in both memory and speed. Furthermore, we will show later that DPW maintains one of the main computational advantages of PW basis, that the matrix elements of the Laplacian operator can be evaluated using Fast Fourier Transform (FFT).

### 3.6 Nonuniform Fast Fourier Transform

A periodic function can be expanded as Fourier series $F(\mathbf{x}) = \sum_{\mathbf{G}} \tilde{F}_{\mathbf{G}} e^{i\mathbf{G}^\top \mathbf{x}}$. Type-2 nonuniform FFT (NUFFT) can be used to efficiently evaluate $F(\mathbf{x})$ on *nonuniform* real space grids $\{\mathbf{x}_i\}$ in quasi-linear time by spreading coefficients to an oversampled uniform grid with a smooth kernel, applying standard FFTs, then interpolating back. We use FINUFFT Barnett (2020), which provides high-accuracy type-2 (uniform $\to$ nonuniform) transforms with rigorous aliasing control, achieving near $O(M \log M)$ complexity where $M$ is the mesh size and near machine precision with modest oversampling and kernel width. In our setting, type-2 NUFFT are used to evaluate periodic functions on distorted grids $\mathbf{x}_i = f(\boldsymbol{\xi}_i)$.

## 4 Flow distorted Plane Waves

### 4.1 Non-cubic unit cell

The original DPW was defined on cubic unit cells $\Omega = [-\frac{a}{2}, \frac{a}{2}]^3$ with the distortion map $g(\boldsymbol{\xi}) = \mathbf{x}$. Here we extended DPW to arbitrary unit cell $\Omega_A$ with cell vector $\mathbf{A}$ by composing the following linear transformation $T : \Omega \to \Omega_A$ to the bijection $g : \Omega \to \Omega$ on the cubic unit cell $\Omega = [-\pi, \pi]^3$:

$$\mathbf{r} = T(\mathbf{x}) = \mathbf{A} \frac{\mathbf{x}}{2\pi}, \quad \mathbf{x} = T^{-1}(\mathbf{r}) = 2\pi \mathbf{A}^{-1} \mathbf{r} = \mathbf{B}^\top \mathbf{r}. \tag{4}$$

Then $f = T \circ g$ is a bijection from the *parameter space* $\Omega$ to the *physical space* $\Omega_A$. Note that the cell vector and reciprocal cell vector of the parameter space cell $\Omega$ are $2\pi\mathbf{I}$ and $\mathbf{I}$ respectively. From now on we will write $J = \frac{\partial r^i}{\partial \xi^\alpha}, J_g = \frac{\partial x^i}{\partial \xi^\alpha}$. Since $\frac{\partial r^i}{\partial x^j} = |A|(2\pi)^{-3} = \frac{\Omega_A}{\Omega}$, we have $|J| = \frac{\Omega_A}{\Omega}|J_g|$.

## 4.2 BLOCH PHASE FACTOR

The original DPW only considers finite systems. Here we extend DPW to periodic systems. First note that in the physical space, the k-points $\mathbf{k}^A$ depend on the cell vector $\mathbf{A}$ of the physical unit cell $\Omega_A$. Naively applying Bloch's theorem yields the following orbital

$$\psi_{n\mathbf{k}}(\mathbf{r}) = e^{\mathrm{i}(\mathbf{k}^A)^\top \mathbf{r}} \sum_{\mathbf{G}} c_{n\mathbf{k}\mathbf{G}} \phi_{\mathbf{G}}(\mathbf{r}) = \sum_{\mathbf{G}} c_{n\mathbf{k}\mathbf{G}} \frac{1}{\sqrt{\Omega}} |J|^{-\frac{1}{2}} \exp\left[\mathrm{i}\mathbf{G}^\top f^{-1}(\mathbf{r}) + \mathrm{i}(\mathbf{k}^A)^\top \mathbf{r}\right]. \quad (5)$$

However, with this choice, the Hamiltonian basis with different $\mathbf{k}^A$ are no longer automatically orthogonal due to the distortion $f$, since if $f$ is not identity, then

$$\left\langle \mathbf{k}^A, \mathbf{G} \middle| \mathbf{k}'^A, \mathbf{G}' \right\rangle = \frac{1}{N_k \Omega} \int_{N_k \Omega} \mathrm{d}^3 \xi \exp\left[\mathrm{i}(\mathbf{G}' - \mathbf{G})\boldsymbol{\xi} + \mathrm{i}(\mathbf{k}'^A - \mathbf{k}^A)^\top f(\boldsymbol{\xi})\right] \neq \delta_{\mathbf{G}\mathbf{G}'} \delta_{\mathbf{k}\mathbf{k}'}. \quad (6)$$

The implication is that one would need to orthogonalize over $\mathbf{k}^A$ as well, instead of only over bands, which would be expensive. Therefore we propose to use the phase shift factor $\exp\left[\mathrm{i}\mathbf{k}^\top f^{-1}(\mathbf{r})\right]$ instead, where $\mathbf{k} = \mathbf{B}^{-1}\mathbf{k}^A$. Since $g$ is a bijection over $\Omega$, we have the decomposition $g = \mathrm{Id} + g_p, g^{-1} = \mathrm{Id} + (g^{-1})_p$ where $g_p$ and $(g^{-1})_p$ are periodic over $\Omega$. Let $g_p(\boldsymbol{\xi}) = \delta\boldsymbol{\xi}$, then $g(\boldsymbol{\xi}) = \boldsymbol{\xi} + \delta\boldsymbol{\xi} = \mathbf{x}$. Now

$$\boldsymbol{\xi} = g^{-1}(\mathbf{x}) = \mathbf{x} + (g^{-1})_p(\mathbf{x}) \quad \Rightarrow \quad (g^{-1})_p(\mathbf{x}) = -\delta\boldsymbol{\xi} = -g_p(\boldsymbol{\xi}) = -g_p(g^{-1}(\mathbf{x})). \quad (7)$$

Now for every point $\mathbf{r}$ in the physical unit cell $\Omega_A$, we have $f^{-1}(\mathbf{r}) = g^{-1}(\mathbf{B}^\top \mathbf{r}) = \mathbf{B}^\top \mathbf{r} + (g^{-1})_p(\mathbf{B}^\top \mathbf{r})$. This means that

$$\exp\left[\mathrm{i}\mathbf{k}^\top f^{-1}(\mathbf{r})\right] = \exp\left[\mathrm{i}(\mathbf{k}^A)^\top \mathbf{r}\right] \exp\left[\mathrm{i}\mathbf{k}^\top (g^{-1})_p(\mathbf{B}^\top \mathbf{r})\right]. \quad (8)$$

The periodic part $\exp\left[\mathrm{i}\mathbf{k}^\top (g^{-1})_p(\mathbf{B}^\top \mathbf{r})\right]$ can be absorbed into the periodic part $u_{n\mathbf{k}^A}$, which amounts to using a different basis to expand each $u_{n\mathbf{k}^A}$. So the orbitals still satisfy the Bloch theorem:

$$\psi_{n\mathbf{k}}(\mathbf{r}) = e^{\mathrm{i}(\mathbf{k})^\top \mathbf{r}} \sum_{\mathbf{G}} c_{n\mathbf{k}\mathbf{G}} \left[\phi_{\mathbf{G}}(\mathbf{r})e^{\mathrm{i}\mathbf{k}^\top (g^{-1})_p(\mathbf{B}^\top \mathbf{r})}\right] = \sum_{\mathbf{G}} c_{n\mathbf{k}^A \mathbf{G}} \phi_{\mathbf{G}+\mathbf{k}}(\mathbf{r}). \quad (9)$$

And it is easy to see that now the Hamiltonian basis between different $\mathbf{k}^A$ is again automatically orthogonal. Furthermore, since this is still a valid parameterization of the Bloch state, we still have k-space decoupling as described in section A.4. Due to the Bloch phase choice $e^{\mathrm{i}\mathbf{k}^\top f^{-1}(\mathbf{r})}$ with $\mathbf{k} = \mathbf{B}^{-1}\mathbf{k}^A$, all physical-space phase factors are handled by evaluating at $f^{-1}(\mathbf{r})$. Consequently, for the remainder of the paper (unless stated otherwise), we drop the superscript and use $\mathbf{G}$ and $\mathbf{k}$ exclusively for wavevectors defined on the cubic parameter cell $\Omega$. Physical-space quantities appear only through $f$ or $f^{-1}$.

## 4.3 BIJECTION ON TORUS

DPW is entirely defined by the bijection on tori $g : \Omega \to \Omega$. Flow distorted plane wave (FDPW) is a DPW where the bijection $g_\theta : \Omega \to \Omega$ is constructed with periodic flow on tori $\Omega$. To create bijection on the parameter space 3-torus $\Omega$, we used the following autoregressive construction:

$$\begin{aligned}
\boldsymbol{\xi}' &= g(\boldsymbol{\xi}; \boldsymbol{\theta}), \quad \boldsymbol{\theta} = (\boldsymbol{\theta}_1, \boldsymbol{\theta}_2, \boldsymbol{\theta}_3) \\
\xi_1' &= g_1(\xi_1; \boldsymbol{\theta}_1) = \texttt{CircularFlow}(\xi_1; \boldsymbol{\theta}_1) \\
\xi_2' &= g_2(\xi_2; \xi_1', \boldsymbol{\theta}_2) = \texttt{CircularFlow}(\xi_2; \texttt{MLP}(\texttt{FF}_N(\xi_1'); \boldsymbol{\theta}_2)) \\
\xi_3' &= g_3(\xi_3; \xi_1', \xi_2', \boldsymbol{\theta}_3) = \texttt{CircularFlow}(\xi_3; \texttt{MLP}(\texttt{FF}_N([\xi_1', \xi_2']); \boldsymbol{\theta}_3))
\end{aligned} \quad (10)$$

where $\texttt{CircularFlow}$ refers to a normalizing flow on $\mathbb{S}^1$, $\texttt{MLP}$ refers to the multi-layer perceptron conditioner, which maps previous dimensions to the flow parameter of the bijection for the current dimension, and $\texttt{FF}_N$ is the fourier features $\texttt{FF}_N(x) = \begin{bmatrix} \cos x & \cos 2x & \dots & \cos 2^N x \\ \sin x & \sin 2x & \dots & \sin 2^N x \end{bmatrix}$ which makes the conditioner MLP a periodic function. In this paper, we use rational-quadratic splines for $\texttt{CircularFlow}$ (section 3.4). The conditioner uses the transformed variable $\boldsymbol{\xi}'$ as input so that the

map is invertible. The advantage of such an autoregressive construction is that the log determinant of the Jacobian $J_f$ can be calculated efficiently since it is triangular.

A single application of the above 3-torus bijection is not expressive enough to express arbitrary distortion. For example, in equation 10, the bijection on $\xi_1$ is unconditioned, so for all $\boldsymbol{\xi}$, the distortion on the first dimension $\xi_1$ will be the same. To ensure sufficient conditioning, we need to apply the above 3-torus bijection multiple times, and permute the dimensions after each bijection. Therefore, we construct the 3-torus bijection as follows

$$g_\theta = g_{\boldsymbol{\theta}(L)} \circ \cdots \circ g_{\boldsymbol{\theta}(2)} \circ \sigma_2 \circ g_{\boldsymbol{\theta}(1)} \circ \sigma_1 \tag{11}$$

where $\sigma_i$ is the permutation map on the input dimensions, where the permutation is taken from one of the six permutations of the set $\{1, 2, 3\}$.

Finally, the affine transformation $T$ (Equation 4) is applied so that $f = T \circ g$ is a bijection from $\Omega$ to $\Omega_A$. We choose the base distribution $p_0$ the uniform distribution $p_0(\boldsymbol{\xi}) = \frac{1}{\Omega} = \frac{1}{8\pi^3}$, so the pullback of $p_0$ to physical space is given by

$$p_\theta(\mathbf{r}) = \frac{1}{\Omega}|J|^{-1} = \frac{1}{\Omega}\frac{\Omega}{\Omega_A}\big|J_{g^{-1}}(\mathbf{r})\big| = \frac{1}{\Omega_A}|J_g(\boldsymbol{\xi}(\mathbf{r}))|^{-1}. \tag{12}$$

## 4.4 PRESCRIBED DENSITY AND INDUCED DISTORTION

Given an unit cell configuration $\{Z_\ell, \boldsymbol{\tau}_\ell\}_\ell$, we use a prescribed density to induces a distortion with grid density increasing at the rate of $\frac{1}{r}$ in the radial range $\left[\frac{a}{Z_\ell}, \frac{b}{Z_\ell}\right]$ from the nucleus of atom $\ell$, while keeping the grid density elsewhere approximately constant. The aim is to accelerate convergence in terms of the number of basis required, since most of the high-frequency component comes from the core orbitals. Following Lindsey & Sharma (2024), we used the following *unnormalized* prescribed density:

$$\rho^{\text{prescribed}}(\mathbf{r}) = \sum_\ell \left[\text{erf}\left(\frac{Z_\ell}{a_\ell}|\mathbf{r} - \boldsymbol{\tau}_\ell|\right) - \text{erf}\left(\frac{Z_\ell}{b_\ell}|\mathbf{r} - \boldsymbol{\tau}_\ell|\right)\right]/|\mathbf{r} - \boldsymbol{\tau}_\ell| + c, \tag{13}$$

where $a_\ell, b_\ell, c$ are hyperparameters. To match the flow density $p_\theta$ to the prescribed density, we minimize the Kullback-Leibler (KL) divergence between the two, where we exploit the fact that $p_\theta$ can be sampled easily:

$$\arg\min_\theta \text{KL}(p_\theta \,||\, \rho^{\text{prescribed}}) = \mathbb{E}_{p_\theta}[\log p_\theta(\mathbf{r}) - \log \rho^{\text{prescribed}}(\mathbf{r})]. \tag{14}$$

To improve the regularity of the grid, we regularize the KL objective with the following elastic energy, with an isochoric shear term and a smoothing term

$$E_{\text{elastic}}(\theta) = \mu_{\text{shear}}\mathbb{E}[\text{tr}(g_{\text{iso}}) + \text{tr}(g_{\text{iso}}^{-1}) - 2d] + \mu_{\text{smooth}}\mathbb{E}[\text{tr}(g)], \tag{15}$$

where $g_{\text{iso}}^{\alpha\beta} = g^{\alpha\beta}|g|^{1/d}$ and $g^{\alpha\beta} = (J^{-1})_i^\alpha(J^{-1})_i^\beta$ is the inverse metric tensor. Figure 4.4 shows a $32^3$ distorted grid for diamond with $\mu_{\text{shear}} = \mu_{\text{smooth}} = 0.005$, $a = 0.1$, $b = 4$, $c = 0.01$. We minimize the objective with AdamW Loshchilov & Hutter (2019) with a learning rate of 0.0002.

## 4.5 COMPARISON TO ANALYTICAL DISTORTION

An alternative way to create the distortion is to use fixed analytical functions to construct the bijection $g$, as done in previous work like Gygi (1995). Specifically, in Gygi's paper, the inverse map is written as an identity plus a sum of isotropic radial bumps around each atom with a small number of parameters per species tuned on one or two reference structures and then transferred to other environments. This construction restricts the map to superpositions of isotropic, atom-centered deformations and requires explicit lattice sums to enforce periodicity. In contrast, our prescribed density approach fits a bijective periodic flow to a "design" density, so no lattice sum is required and we can use arbitrary design density. This allows us to capture highly anisotropic features (covalent lobes, surfaces, low symmetry bonding). The same mechanism can also be reused for downstream tasks such as density inversion or surface specific distortions. Furthermore, all geometric quantities needed for the Laplace-Beltrami operator, the contracted connection $A = \partial_\beta \log|J|$ and the inverse metric $g$, can be calculated efficiently via AD. Log Jacobian determinant $\log|J|$ is readily available and cheap to calculate for all normalizing flows since it is required for evaluating probability, and its derivatives can be obtained via AD. $g$ can be obtained by applying AD to the flow bijector, which is also straightforward.

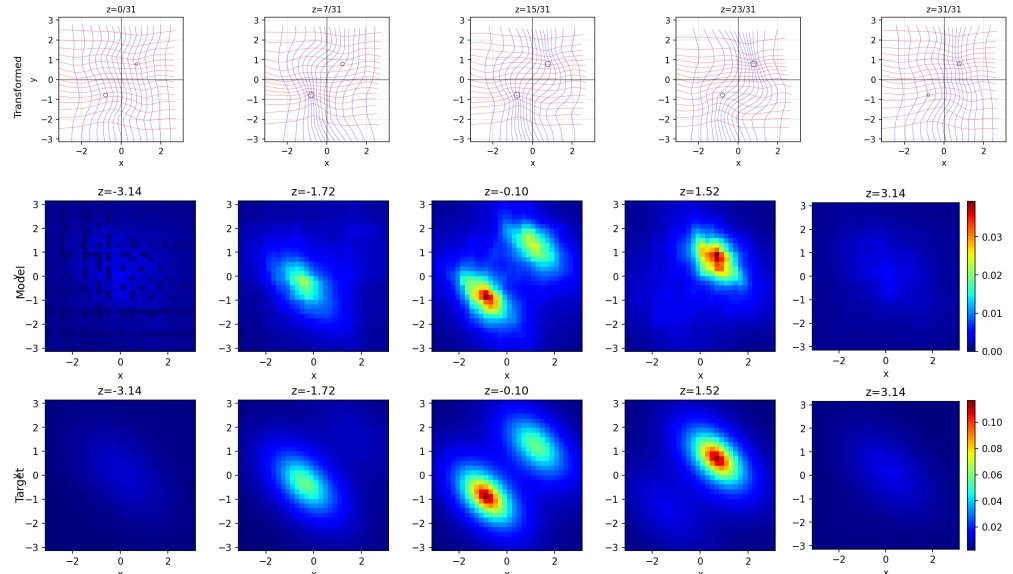

Figure 1: Distorted grid and prescribed density slices $p_\theta$ (middle) vs. target $\rho^{\text{prescribed}}/Z$ (bottom) at selected $z$ in reduced coordinates $\mathbf{B}^\top \mathbf{r}$. Partition $Z$ is estimated with the flow $p_\theta$. Circle size indicates proximity of nuclei to the shown $xy$-plane.

### 4.6 METRIC-WEIGHTED DENSITY MATRIX

We define a metric-weighted version of $u_{n\mathbf{k}}$, $S_{n\mathbf{k}}(\mathbf{r}) = |J|u_{n\mathbf{k}}(\mathbf{r})$, whose evaluation on the distorted grid is given by $\mathbf{S}_{n\mathbf{k}} = \frac{N}{\sqrt{\Omega}}\text{FFT}^{-1}(\mathbf{c}_{n\mathbf{k}})$. Similarly, we define $S_{\mathbf{k},nm} = S_{n\mathbf{k}}^*(\mathbf{r})S_{m\mathbf{k}}(\mathbf{r})$, which is the metric-weight version of the band-resolved density matrix $\Gamma_{\mathbf{k},nm}(\mathbf{r}) = u_{n\mathbf{k}}^*(\mathbf{r})u_{m\mathbf{k}}(\mathbf{r})$. Since the density is given by $\rho(\mathbf{r}) = \sum_{\mathbf{k}} \text{tr}[\mathbf{F}_{\mathbf{k}}\Gamma_{\mathbf{k},nm}(\mathbf{r})]$, we also have the metric-weighted density $S(\mathbf{r}) = |J|\rho(\mathbf{r}) = \sum_{\mathbf{k}} \text{tr}[\mathbf{F}_{\mathbf{k}}S_{\mathbf{k},nm}(\mathbf{r})]$. For local operator $O(\mathbf{r})$, its matrix element is *diagonal*: $O_{\mathbf{G}'\mathbf{G}} = \delta_{\mathbf{G}'\mathbf{G}}O_{\mathbf{0}\mathbf{G}}$. Let $\mathbf{O}, \mathbf{S}_{\mathbf{k},nm} \in \mathbb{C}^N$ are the evaluation of $O(\mathbf{r})$ and $S_{\mathbf{k},nm}(\mathbf{r})$ on the distorted grid $\{\mathbf{r}_i = f(\boldsymbol{\xi}_i)\}_{i=1}^N$, then we have (see Appendix F)

$$\langle\psi_{n\mathbf{k}}|\hat{O}|\psi_{m\mathbf{k}}\rangle \simeq \frac{\Omega}{N}\mathbf{S}_{\mathbf{k},nm}^\dagger\mathbf{O} = N[\text{FFT}^{-1}(\mathbf{c}_{n\mathbf{k}})^* \odot \text{FFT}^{-1}(\mathbf{c}_{m\mathbf{k}})]^\dagger\mathbf{O}. \tag{16}$$

### 4.7 KINETIC OPERATOR

The kinetic energy is $E_{\text{kin}} = \sum_{n\mathbf{k}} f_{n\mathbf{k}} \langle\psi_{n\mathbf{k}}|\hat{T}|\psi_{n\mathbf{k}}\rangle = \sum_{\mathbf{G}'\mathbf{G}} c_{n\mathbf{k}\mathbf{G}'}^* c_{m\mathbf{k}\mathbf{G}}T_{\mathbf{k},\mathbf{G}'\mathbf{G}}$, where $\hat{T} = -\frac{1}{2}\Delta$ in atomic unit. Unlike in PW basis, $\hat{T}$ is *not diagonal* under FDPW basis (see section 3.3), and we cannot use the Eq. 16 from the last section. However, we can still avoid forming the full $T_{\mathbf{k},\mathbf{G}'\mathbf{G}}$ matrix and evaluate $E_{\text{kin}}$ efficiently using FFT under the FDPW basis. Denoting rescaled contracted connection as $A'_\beta = \frac{1}{2}\Gamma^\alpha_{\alpha\beta} = \frac{1}{2}\partial_\beta \log|J|$ (see Appendix E.6). On the half-density bundle, the pullback of $\Delta$ has the following symmetric "minimal coupling" form

$$\int_{\Omega_A}(|J|^{-\frac{1}{2}}\Phi^*)\Delta_{\mathbf{r}}(|J|^{-\frac{1}{2}}\Psi)\mathrm{d}^3r = -\int_\Omega[(-A'_\alpha + \partial_\alpha)\Phi^*]g^{\alpha\beta}[(-A'_\beta + \partial_\beta)\Psi]\,\mathrm{d}^3\xi, \tag{17}$$

analogous to the Euclidean Laplacian after integration by parts (see Appendix E.9). When $\Phi(\boldsymbol{\xi}) = \Psi(\boldsymbol{\xi}) = \sum_{\mathbf{G}} c_{\mathbf{G}}e^{i(\mathbf{G}+\mathbf{k})^\top\boldsymbol{\xi}}$, the above formula becomes a quadratic forms which can be evaluated with two inverse FFTs, and a few point-wise multiplications on the distorted grid (see Appendix G for derivation):

$$\langle\psi_{n\mathbf{k}}|\hat{T}|\psi_{m\mathbf{k}}\rangle = \frac{1}{2}\frac{\Omega}{N}\sum_{i=1}^N \mathbf{W}_{\alpha,n\mathbf{k}}^*(\boldsymbol{\xi}_i)g^{\alpha\beta}(\boldsymbol{\xi}_i)\mathbf{W}_{\beta,m\mathbf{k}}(\boldsymbol{\xi}_i), \tag{18}$$

where $\mathbf{W}_{\beta,n\mathbf{k}}(\boldsymbol{\xi}) = \frac{N}{\sqrt{\Omega}}[-A'_\beta(\boldsymbol{\xi})\text{FFT}^{-1}(\mathbf{c}_{n\mathbf{k}}) + \text{FFT}^{-1}(i(\mathbf{G}+\mathbf{k})\mathbf{c}_{n\mathbf{k}})] \in \mathbb{C}^{N_{\text{basis}}}$.

### 4.8 POTENTIAL OPERATORS

Given two charge densities $\rho_1, \rho_2 : \Omega_A \to \mathbb{R}^+$ that are periodic and charge-neutral (i.e. zero-mean over $\Omega_A$), we write the Coulomb interaction energy between them as $(\rho_1|\rho_2) = \int_{\Omega_A} V_{\rho_1}(\mathbf{r})\rho_2(\mathbf{r})\,\mathrm{d}\mathbf{r}$ where $V_{\rho_1}(\mathbf{r}) = \int_{\mathbb{R}^3} \frac{1}{\|\mathbf{r}-\mathbf{r}'\|}\rho_1(\mathbf{r}')\,\mathrm{d}\mathbf{r}'$ is the Coulomb potential generated by $\rho_1$, and $((\rho_1)) = (\rho_1|\rho_1)$. Denote the atomic point charge as $\rho^{\mathrm{atom}}(\mathbf{r}) = -\sum_\ell Z_\ell \delta(\mathbf{r} - \boldsymbol{\tau}_\ell)$. In periodic system, the total classical potential energies $\frac{1}{2}((\rho + \rho^{\mathrm{atom}}))$ is only conditionally convergent. We need to split it into three convergent series that consist of Coulomb interaction between charge-neutral densities (see Appendix J):

$$\frac{1}{2}((\rho + \rho^{\mathrm{atom}})) = \underbrace{\frac{1}{2}((\rho + \rho^+))}_{\text{Hartree}} + \underbrace{(\rho + \rho^+|\rho^{\mathrm{atom}} + \rho^-)}_{\text{External}} + \underbrace{\frac{1}{2}((\rho^{\mathrm{atom}} + \rho^-))}_{\text{Nucleus}}, \tag{19}$$

where $\rho^\pm(\mathbf{r}) = \mp Z_{\mathrm{tot}}/\Omega$ are uniform background charges. In DFT, the exchange and interaction are modelled by the XC functional $E_{\mathrm{XC}}[\rho] = \int_{\Omega_A} \varepsilon_{\mathrm{XC}}[\rho]\rho\,\mathrm{d}\mathbf{r}$ where $\varepsilon_{\mathrm{XC}}$ is the per particle energy density, so the total potential energy is the above classical term plus $E_{\mathrm{XC}}[\rho]$. In PW basis, all these energy terms have analytical formula as the Coulomb potential $\frac{1}{r}$ has analytic Fourier transform (see Appendix H). For FDPW, we do not have an analytical formula for $\langle \phi_{\mathbf{G}} | \hat{V} \rangle$. If local XC functional is used, then the effective potential operator $\hat{V}_{\mathrm{eff}}[\rho] = V_H[\rho] + V_{\mathrm{ext}} + \varepsilon_{\mathrm{XC}}[\rho]$ is local. Let $\mathbf{V}_{\mathrm{eff}}[\rho] \in \mathbb{R}^N$ be the evaluation of $V_{\mathrm{eff}}[\rho]$ on the distorted grid $\{\mathbf{r}_i = f(\boldsymbol{\xi}_i)\}_{i=1}^N$, by eq. 16 we have

$$\langle \psi_{n\mathbf{k}} | \hat{V}_{\mathrm{eff}}[\rho] | \psi_{m\mathbf{k}} \rangle \simeq \frac{\Omega}{N} \mathbf{S}_{\mathbf{k},nm}^\dagger \mathbf{V}_{\mathrm{eff}}[\rho]. \tag{20}$$

All we need to do is to have convergent real-space expressions for $V_H[\rho]$ and $V_{\mathrm{ext}}$ and evaluate them on the distorted grid. Note that semi-local XC can also be computed similarly by applying the pullback rule of the gradient $\nabla$ operator.

### 4.9 HARTREE POTENTIAL

Since the scaled Coulomb kernel $-\frac{1}{4\pi r}$ is the Green's function of the 3D Laplacian operator $\nabla^2$, one can solve for the Hartree potential $V_H$ via the Poisson equation with periodic boundary condition:

$$V_H[\rho] := V_\rho = \rho \star \frac{1}{r} \quad \Rightarrow \quad -\nabla^2 V_H[\rho] = 4\pi(\rho - \overline{\rho}). \tag{21}$$

We will suppress the $\rho$ dependence of $V_H$ from now on when it is not ambiguous. Using the pullback of Laplace-Beltrami operator $|J|^{-1}\partial_\alpha \left(|J|g^{\alpha\beta}\partial_\beta\phi\right)$ on scalar field $\phi$ and multiply both side by $|J|$, we get the conservative form Poisson equation in $\boldsymbol{\xi}$-space:

$$-\partial_\alpha \left(|J|g^{\alpha\beta}\partial_\beta V_H\right) = 4\pi|J|(\rho - \overline{\rho}) = 4\pi(S - \overline{S}), \tag{22}$$

where $S = |J|\rho$ is the metric-weighted density we defined in section 4.6. We apply Galerkin approximation with FDPW basis to this equation, where $V_H$ is parameterized as its value over the uniform $\xi$-space grid, and the matrix-vector-product $-\partial_\alpha \left(|J|g^{\alpha\beta}\partial_\beta V_H\right)$ can be performed in Fourier space, since $\langle \phi_{\mathbf{G}} | \partial_\alpha V_H(\boldsymbol{\xi}_i) \rangle = \mathrm{i}\mathbf{G}\tilde{V}_H(\boldsymbol{\xi}_i)$ hence we parameterize $\mathbf{V}_H$ where $(\mathbf{V}_H)_{[i]} = V_H(\boldsymbol{\xi}_i)$, and $\tilde{\mathbf{V}}_H = \mathtt{FFT}(\mathbf{V}_H)$. Under this discretization the operator $-\partial_\alpha \left(|J|g^{\alpha\beta}\partial_\beta V_H\right)$ becomes a matrix-vector-product in the Fourier space

$$-\mathrm{i}\mathbf{G}_\alpha \odot \mathbf{J} \odot \mathbf{g}^{\alpha\beta} \odot \mathtt{FFT}^{-1}[\mathrm{i}\mathbf{G}_\beta \odot \tilde{\mathbf{V}}_H]. \tag{23}$$

where $\odot$ is Hadamard product, $\mathbf{g}, \mathbf{J}$ are the evaluation of $g, |J|$ on the distorted grid. We use a preconditioned conjugate gradient (PCG) to solve for $\tilde{\mathbf{V}}_H$ given $\rho$, where we used a diagonal spectral preconditioner $M^{-1}(\mathbf{G}) = (\tilde{G}^{\alpha\beta}G_\alpha G_\beta)^{-1}$, where $\tilde{G}^{\alpha\beta} = \langle sg^{\alpha\beta} \rangle$, $s = e^{-m}|J|$ and $m = \langle \log|J| \rangle$. Here $\langle \cdot \rangle$ means the average over all entries in the matrix.

#### 4.9.1 EXTERNAL AND NUCLEUS POTENTIAL

The external potential $V_{\mathrm{ext}}$ is generated by point charges $\rho^{\mathrm{atom}}$, so

$$V_{\mathrm{ext}}(\mathbf{r}) = \rho^{\mathrm{atom}} \star \frac{1}{r} = -\sum_\ell Z_\ell \int_{\mathbb{R}^3} \mathrm{d}\mathbf{r}' \, \delta(\mathbf{r}' - \boldsymbol{\tau}_\ell)\frac{1}{\|\mathbf{r}-\mathbf{r}'\|} = -\sum_{\ell\mathbf{R}} Z_\ell \frac{1}{\|\mathbf{r} - \boldsymbol{\tau}_\ell - \mathbf{R}\|}. \tag{24}$$

**Algorithm 1** Ground-state search with FDPW

**Input:** cell $\{Z_\ell, \boldsymbol{\tau}_\ell\}_\ell$, FFT size $N$;
Initialize $\theta$, $\mathbf{w} \in \mathbb{C}^{N_{\text{band}} \times N_k \times N_{\text{basis}}}$, Hartree $\mathbf{V}_H \in \mathbb{R}^N$, and uniform grid $\{\boldsymbol{\xi}_i\}_{i=1}^N$ on $\Omega$;
Fit flow $p_\theta$ to target $\rho_{\text{target}}$ (eq. 14);
Compute and cache $f(\boldsymbol{\xi}_i)$, $g^{\alpha\beta}(\boldsymbol{\xi}_i)$, $A'_\beta(\boldsymbol{\xi}_i)$, and $V_{\text{ext}}(f(\boldsymbol{\xi}_i))$;
**repeat**
    minimize $L_{\text{gs}}(\mathbf{w})$ (eq. 26) for 1 step;
    PCG solve for $\mathbf{V}_H$ (section 4.9) for $n_{\text{iter}}$ steps;
**until** converged

**Algorithm 2** Band-structure evaluation with FDPW

**Input:** cell $\{Z_\ell, \boldsymbol{\tau}_\ell\}_\ell$, FFT size $N$, $\theta$, $\mathbf{V}_H$, $g^{\alpha\beta}(\boldsymbol{\xi}_i)$, $A'_\beta(\boldsymbol{\xi}_i)$, $f(\boldsymbol{\xi}_i)$, $V_{\text{ext}}(f(\boldsymbol{\xi}_i))$, converged $\mathbf{w}^\star$ from ground state search;
**for** $\mathbf{k}_i$ in k-path **do**
    Initialize $\mathbf{w}_{\mathbf{k}_i} \in \mathbb{C}^{N_{\text{band}} \times N_{\text{basis}}}$ to $\mathbf{w}^\star$ or $\mathbf{w}_{\mathbf{k}_{i-1}}$ if $i > 1$
    **repeat**
        minimize $L_{\mathbf{k}_i}$ for 1 step (eq. 26)
    **until** converged
    diagonalize $H_{\mathbf{k}_i, nm}(\mathbf{c}_{\mathbf{k}_i}, \boldsymbol{\rho}^\star)$ to obtain $\{\epsilon_n(\mathbf{k}_i)\}_{n=1}^{N_{\text{band}}}$
**end for**

where $\mathbf{R}$ run over the Bravais lattice. Two issues arise: (i) $\rho^{\text{atom}}$ is not neutral, so the direct energy diverges; (ii) the real-space sum converges slowly with cell size. We use Ewald summation, splitting $V(r) = 1/r$ into a smooth long-range part $V_\eta(r) = Z \operatorname{erf}(\eta r)/r$ and a short-range part with $\operatorname{erfc}(\eta r)$ decay, yielding (see Appendix L):

$$V_{\text{ext}}(\mathbf{r}) = \sum_\ell \{ \sum_{\mathbf{R}} [V - V_\eta](\mathbf{r} - \boldsymbol{\tau}_\ell - \mathbf{R}; Z_\ell) + \sum_{\mathbf{G} \neq \mathbf{0}} \tilde{V}_\eta(\mathbf{G}; Z_\ell) e^{i\mathbf{G} \cdot (\mathbf{r} - \boldsymbol{\tau}_\ell)} \}. \tag{25}$$

This reduces cutoffs and allows local pseudopotentials by replacing $V$ with $V_{\text{loc}}$ in the real-space term. In our experiments, we use the analytical norm-conserving (ANC) regularized potential Gygi (2023). The long-range reciprocal sum is evaluated on distorted grids via type-2 NUFFT (section 3.6). The nucleus-nucleus energy $E_{\text{nuc}}$ has a similar form and also benefits from Ewald summation for rapid convergence (Appendix M).

# 5 ALGORITHM

## 5.1 GROUND-STATE SEARCH

We solve the Kohn-Sham equation via direction minimization (see Appendix A.1) which minimizes the Rayleigh quotient of the Kohn-Sham Hamiltonian matrix $H_{\mathbf{k}, nm}(\mathbf{c}, \boldsymbol{\rho}) = \langle \psi_{n\mathbf{k}} | \hat{T} + \hat{V}_{\text{eff}}[\rho] | \psi_{m\mathbf{k}} \rangle$, where calculation of kinetic contribution follows section 4.7, the potential contribution follows section 4.8, and the evaluate of density on distorted grid $\boldsymbol{\rho} \in \mathbb{R}^{N_{\text{basis}}}$ is given by $\boldsymbol{\rho}(\mathbf{c}) = \mathbf{J}^{-1} \operatorname{tr}(\mathbf{F}_\mathbf{k} \mathbf{S}_{\mathbf{k}, nm}(\mathbf{c}))$ follows section 4.6:

$$\min_{\mathbf{w} \in \mathbb{C}^{N_{\text{band}} \times N_k \times N_{\text{basis}}}} L_{\text{gs}}(\mathbf{w}) = \sum_\mathbf{k} \operatorname{tr}[\mathbf{F}_\mathbf{k} H_{\mathbf{k}, nm}(\mathbf{c}, \texttt{SG}(\boldsymbol{\rho}(\mathbf{c})))], \quad \mathbf{c} = \texttt{QR}(\mathbf{w}) \tag{26}$$

where a QR retraction for the orthonormal constraint $\langle \psi_{n\mathbf{k}} | \psi_{m\mathbf{k}} \rangle = \mathbf{c}_{n\mathbf{k}}^\dagger \mathbf{c}_{m\mathbf{k}} = \delta_{nm}$ is used to map the unconstrained parameter $\mathbf{w}$ to the FDPW coefficients $\mathbf{c}$, and we put the stop gradient op $\texttt{SG}$ around $\rho$ when computing $\hat{V}_{\text{eff}}[\rho]$, which effectively converts the nonlinear eigenvalue problem into a linear eigenvalue problem. Algorithm 1 describes the full routine where the convergence is determined by checking whether the standard deviation of the total energy of the current ansatz is below a set threshold. Note that the PCG solver can be amortized over the minimization of the main objective, and in practice, setting $n_{\text{iter}} = 1$ suffices (see appendix R).

## 5.2 BAND STRUCTURE CALCULATION

Band structure calculation is carried out after the ground-state search. We follow the usual non-self-consistent-field (NSCF) procedure where the $V_{\text{eff}}$ is fixed by fixing the density to the ground state energy calculated using a coarse k-mesh. The Hamiltonian trace is minimized at each $\mathbf{k}$ along the k-path with warm start, similar to Tianbo Li (2024). The loss is the same as eq. 26 except for that we do not apply the occupation $\mathbf{F}_\mathbf{k}$ and the $V_{\text{eff}}[\rho^\star]$ uses the ground state density $\rho^\star$ computed from the ground state search: $L_\mathbf{k}(\mathbf{w}_\mathbf{k}) = \operatorname{tr}[H_{\mathbf{k}, nm}(\texttt{QR}(\mathbf{w}_\mathbf{k}), \boldsymbol{\rho}^\star)]$, where $\mathbf{w}_\mathbf{k} \in \mathbb{C}^{N_{\text{band}} \times N_{\text{basis}}}$ is the slice of $\mathbf{w}$ at $\mathbf{k}$. Algorithm 2 describes the full routine. The main objective is minimized with AdamW in both the ground state search and band structure calculation.

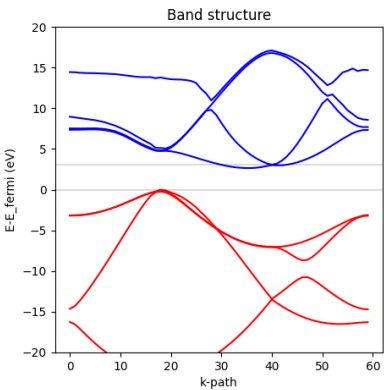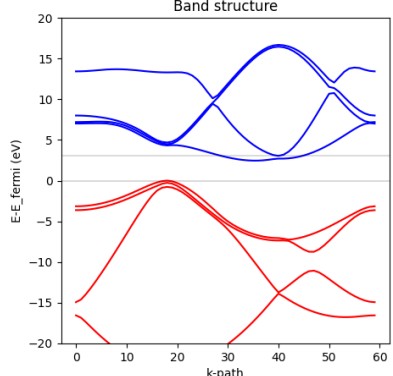

Figure 2: Diamond band structures computed with standard PW (left) with $N = 128$ and FDPW (right) with $N = 64$. FDPW preserves dispersion features while using a lower grid size $N$.

### 5.3 GROUND-STATE SEARCH FOR FINITE SYSTEM

For finite systems, a cubic unit cell $\Omega_A = [-\frac{a}{2}, \frac{a}{2}]^3$ with large enough $a$ is used to create a vacuum around the target system. The ground state search algorithm is identical to the one described in section 5.1, except for the following modification: (a) there is no $\mathbf{k}$; (b) we use the free-space gauge for all potentials, i.e. we explicitly set the $\mathbf{G} = 0$ term for $V_H$ (Appendix O) and for $V_{\text{ext}}$ (Appendix L); (c) we do not need to do Ewald summation for $E_{\text{nuc}}$ as in section 4.9.1.

## 6 EXPERIMENTS

We tested FDPW on both finite (molecular) and periodic systems. All calculations were done in a single Nvidia A100 GPU with 40GB of memory. Hyperparameter setting is at Appendix P.

### 6.1 DIAMOND BAND STRUCTURE

We conduct a $\Gamma$-point only ground state search, followed by the NSCF band structure calculation as outlined in section 5. Figure 2 compares band structures from PW and FDPW; both agree closely along the tested k-path, while FDPW achieves similar accuracy at coarser grids. Table 6.1 further summarizes convergence metrics. Crucially, the FDPW band gap converges to $\sim 3.05$ eV with only $N = 64$, while regular PW needs $N = 128$.

Table 1: Band structure calculation for diamond with LDA. All energies are in eV units.

| Method | $N$ | band gap | L | X | $\Gamma$ | Speed (it/s) $\uparrow$ | Mem. (GBs) $\downarrow$ |
|---|---|---|---|---|---|---|---|
| PW | 128 | 3.05869 | 7.51526 | 3.05869 | 4.79634 | 3.68 | 23.5 |
| | 96 | 3.14795 | 7.61833 | 3.14795 | 4.77946 | 11.38 | 15.4 |
| | 64 | 3.68519 | 7.72595 | 3.68519 | 4.88759 | 44.48 | 1.34 |
| | 48 | 3.93325 | 7.73193 | 3.93325 | 4.83815 | 106.05 | 1.24 |
| | 32 | 4.96448 | 8.17394 | 5.18815 | 4.96448 | 144.49 | 1.15 |
| FDPW | 64 | 3.04445 | 7.19263 | 3.04445 | 4.46958 | 32.78 | 1.20 |
| | 48 | 3.01882 | 7.26194 | 3.01882 | 4.41584 | 108.86 | 1.41 |
| | 32 | 4.22392 | 5.69283 | 6.95550 | 4.35785 | 157.92 | 1.17 |

### 6.2 FINITE SYSTEMS

We follow Gygi (1993) and use a cubic box of length $a = 10$ Bohr. We consider the CO (carbon monoxide) molecule with geometry [[-1.065, 0.0, 0.0], [1.065, 0.0, 0.0]] and compute the ground state with both PW and FDPW. PySCF Sun et al. (2017) was used to compute the reference value. See Appendix Q for details.

## 7 CONCLUSION AND FUTURE WORKS

We introduced flow-distorted plane waves (FDPW): a PW basis composed with a bijective, periodic normalizing flow on the 3-torus. FDPW adapts resolution where needed while preserving PW algebra, k-point orthogonality via a modified Bloch phase, and FFT/NUFFT efficiency. We extended Gygi's DPW to arbitrary lattices and to both finite and periodic settings, and proposed a compact neural parameterization with prescribed-density initialization and regularization. We have demonstrated that, on both molecules and solids, FDPW can effectively lower the grid size required for convergence, and being an *all-electron* method, core electrons can be modeled unlike other PW+pseudopotential framework that only models valence electrons, and no predefined pseudopotential wave are needed since everything can be computed on the fly.

Future work includes joint flow/SCF training, richer Hamiltonians (nonlocal PPs, spin, SOC, hybrid XC), differentiable forces for geometry and MD, and scaling with improved preconditioning and parallelism. Furthermore, the use of neural networks on an adaptive basis enables the training of a "foundation model" for a basis set which can adapt to new geometry without the density fitting steps, thus opening new vistas for ab-initio calculation at the mean-field level.

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

# A DENSITY FUNCTIONAL THEORY

## A.1 SOLVING THE KS EQUATION

In Hohenberg, Kohn, and Sham's density functional theory (DFT) Hohenberg & Kohn (1964); Kohn & Sham (1965), The density of the ground state of an electronic Hamiltonian can be solved via an auxiliary Kohn-Sham (KS) system:

$$\min_{\hat{\gamma}} E[\hat{\gamma}] = \text{tr}\left[\hat{\gamma}\hat{E}[\rho]\right], \quad \hat{E}[\rho] = \hat{T} + \hat{V}_{\text{ext}} + \varepsilon_{\text{Hxc}}[\rho], \tag{27}$$

where $\hat{\gamma} = \sum_n f_n |\psi_n\rangle\langle\psi_n|$ is the one-body reduced density matrix (1-RDM), $f_n \in \{0, 1\}$ is the occupation number, $|\psi_n\rangle$ are orthonormal one-body wavefunctions also known as orbitals, $\varepsilon_{\text{Hxc}}[\rho]$ is the energy density of the Hartree and exchange-correlation (XC) energy, and the density $\rho(\mathbf{r})$ can be obtained by taking the trace in real space of 1-RDM: $\rho(\mathbf{r}) = \langle\mathbf{r}|\hat{\gamma}|\mathbf{r}\rangle = \sum_n f_n |\psi_n(\mathbf{r})|^2$. Under a fixed occupation $\{f_n\}$, the Lagrangian of the above constrained optimization problem is

$\mathcal{L}(\{\psi\}, \lambda) = E[\hat{\gamma}] - \sum_{nm} \lambda_{nm} [\langle \psi_n | \psi_m \rangle - \delta_{nm}]$, whose stationary condition $\frac{\delta \mathcal{L}}{\delta \psi_n^*} = 0$ yields the Kohn-Sham equation (KS)

$$\hat{H}[\rho] |\psi_n\rangle = \varepsilon_n |\psi_n\rangle. \tag{28}$$

where $\varepsilon_n$ are called the KS eigenvalues.

Galerkin approximation can be applied to convert the above infinite-dimensional problem to a finite-dimensional problem. Given a truncated complete basis $\{|\phi_p\rangle\}_{p=1}^{N_{\text{basis}}}$ of size $N_{\text{basis}}$, the orbital $\psi_n$ can be represented as the basis coefficients $\mathbf{c}_n \in \mathbb{C}^{N_{\text{basis}}}$ where $c_{np} = \langle \phi_p | \psi_n \rangle$. Any operator $\hat{O}$ can be represented as a matrix $O_{pq} = \langle \phi_p | \hat{O} | \phi_q \rangle$, and its action on some orbital $\psi_n$ can be calculated as a matrix vector product $\hat{O} |\psi_n\rangle \approx \sum_q O_{pq} c_{nq}$. Specifically, the Hamiltonian matrix $H_{nm}[\rho] = \langle \psi_n | \hat{H}[\rho] | \psi_m \rangle$ can be computed from the matrix element $H_{pq}[\rho] = \langle \phi_p | \hat{H}[\rho] | \phi_q \rangle$ as

$$H_{nm}[\rho] = \sum_{pq} c_{np}^* c_{mq} H_{pq}[\rho]. \tag{29}$$

Usually, the variational problem of minimizing the DFT Lagrangian is solved via applying the Self-Consistent-Field (SCF) iteration. SCF loop iterates between diagonalizing the current Hamiltonian matrix $H_{nm}[\rho^{(t)}] \mapsto \{\psi_n^{(t)}\}$ and update the Hamiltonain with the new density from the eigenfunctions of last step $\{\psi_n^{(t)}\} \mapsto \rho^{(t+1)} \mapsto H_{nm}[\rho^{(t+1)}]$.

Alternatively, one can directly minimize the DFT Lagrangian. The problem can be further transformed into an unconstrained optimization of the total energy $E_{\text{el}}$ as in Li et al. (2023), where the orthogonal constraints are handled by the QR retraction to the Stiefel manifold

$$C = \texttt{Qfactor}(\mathbf{w}) \tag{30}$$

where orthogonal $\mathbf{c}$ is obtained from a skinny QR decomposition of an unconstrained $\mathbf{w}$.

## A.2 BRAVAIS LATTICE

A 3D Bravais lattice is the tilling of the parallelepiped $\Omega$ known as the unit cell, formed by cell vectors $\mathbf{a}_1, \mathbf{a}_2, \mathbf{a}_3 \in \mathbb{R}^3$. We can write the cell vectors more compactly as a matrix $\mathbf{A} = (\mathbf{a}_1 \quad \mathbf{a}_2 \quad \mathbf{a}_3)$. Topologically, the unit cell is a 3-torus due to the periodic boundary condition. The Bravais lattice can be identified with the set of points

$$\mathbf{R}_{\mathbf{m}} = \sum_d m_d \mathbf{a}_d, \quad m_d \in \mathbb{Z}. \tag{31}$$

One can construct 3D periodic functions by specifying their value over the unit cell. Such functions have Fourier series expansion under the PW basis $\frac{1}{\sqrt{\Omega_A}} \exp\left[i\mathbf{G}^\top \mathbf{r}\right]$ where the wavevector $\mathbf{G}$ lies on the reciprocal lattice formed by the reciprocal cell vectors $\mathbf{b}_i$:

$$\mathbf{G}_{\mathbf{n}} = \sum_d n_d \mathbf{b}_d, \quad \mathbf{b}_i = \frac{2\pi}{\Omega_A}(\mathbf{a}_j \times \mathbf{a}_k). \tag{32}$$

Again, we can write the reciprocal cell vectors compactly as $\mathbf{B} = (\mathbf{b}_1 \quad \mathbf{b}_2 \quad \mathbf{b}_3)$, and we have $\mathbf{B} = 2\pi \mathbf{A}^{-\top}$. PW can be identified with a uniform grid of sample points of size $N_1 \times N_2 \times N_3$ over the unit cell $\Omega_A$

$$\mathbf{r}_{\mathbf{n}} = \sum_d \frac{n_d}{N_d} \mathbf{a}_d, \quad n_d \in [-\lfloor (N_d - 1)/2 \rfloor, \lfloor N_d/2 \rfloor] \tag{33}$$

since a periodic function can be fully specified over its values on $\{\mathbf{r}_{\mathbf{n}}\}$ if under frequency representation, its wavevectors lie on the reciprocal lattice of size $N = N_1 \times N_2 \times N_3$.

## A.3 MODELING SOLID-STATE PHYSICS WITH DFT

A crystal can be specified by the cell vectors $\mathbf{a}_i$ of the unit cell, and the atomic configuration $\{Z_\ell, \boldsymbol{\tau}_\ell\}_\ell$ within the unit cell, where $Z_\ell$ is the charge of atom $\ell$ and $\boldsymbol{\tau}_\ell$ is the coordinate. To capture interaction between different translated copies of the unit cells, a finite Bravais lattice with periodic boundary conditions (PBC) is typically used. The finite Bravais lattice is commonly referred to as the *simulation cell*, which we denote as $N_k \Omega_A$ where $\Omega_A$ is the unit cell.

Different from molecular systems, in periodic systems, the potential is periodic since at each location $\mathbf{r}$ within the unit cell, potentials from all translated unit cells are felt. In other words, the tiling periodizes the non-periodic atomic Hartree and external potential. Bloch theorem Bloch (1929) states that, for periodic potential $V(\mathbf{r} + \mathbf{R_m}) = V(\mathbf{r})$, the eigenstates of the Hamiltonian $\hat{H}$ takes the form

$$\psi_{n\mathbf{k}}(\mathbf{r}) = \exp\left[\mathrm{i}\mathbf{k}^\top \mathbf{r}\right] u_{n\mathbf{k}}(\mathbf{r}), \quad \mathbf{k} = \sum_d k_d \mathbf{b}_d \tag{34}$$

where $n$ is the band index and $u_{n\mathbf{k}}$ is a function periodic over the unit cell. Note that the density only depends on $u_{n\mathbf{k}}$, since $\rho(\mathbf{r}) = \sum_{n\mathbf{k}} f_{n\mathbf{k}}|\psi_{n\mathbf{k}}(\mathbf{r})|^2 = \sum_{n\mathbf{k}} f_{n\mathbf{k}}|u_{n\mathbf{k}}(\mathbf{r})|^2$. To make sure that $\psi_{n\mathbf{k}}$ is periodic over the simulation cell of size $M_1 \times M_2 \times M_3$, the k-points $\mathbf{k}$ can only take values from the lattice $k_d = m_d/M_d, m_d \in \mathbb{Z}$. Furthermore, $\mathbf{k}$ within the first Brillouin zone (FBZ), i.e. $m_i \in [-\lfloor (M_i - 1)/2 \rfloor, \lfloor M_i/2 \rfloor]$, gives all unique eigenvalues due to the periodicity in the reciprocal space. All $\mathbf{k}$ within FBZ form a reciprocal lattice with size $N_k = M_1 \times M_2 \times M_3$. Thus, for periodic systems, the KS equation becomes

$$\hat{H}[\rho]\psi_{n\mathbf{k}} = \epsilon_{n\mathbf{k}}\psi_{n\mathbf{k}}. \tag{35}$$

For each $n$, there are distinct energy levels for each $\mathbf{k}$, and the collection $\epsilon_{n\mathbf{k}}$ for fixed $n$ forms a line that is commonly referred to as the $n$-th *band*. Analogous to the HOMO-LUMO gap in molecular systems, the narrowest gap between the highest occupied band and the lowest unoccupied band is referred to as the band gap, which is an important indicator of the electronic conductivity of the system.

With Galerkin approximation, we expand the periodic part of the Bloch state $u_{n\mathbf{k}}$ with a periodic basis $|\phi_n\rangle$ on the unit cell $\Omega_A$, and we have coefficients $c_{nkn} = \langle\phi_n|u_{n\mathbf{k}}\rangle$. The basis used for the Hamiltonian becomes $|\mathbf{k}, n\rangle = \left|e^{\mathrm{i}\mathbf{k}^\top \mathbf{r}}\phi_n\right\rangle$ which is defined on the simulation cell $N_k\Omega_A$, and the Hamiltonian matrix $H[\rho]_{\mathbf{k}n,\mathbf{k}'m} := \langle\mathbf{k}, n|\hat{H}[\rho]|\mathbf{k}', m\rangle$ has size $(N_k \times N_{\text{band}})^2$.

## A.4 K-SPACE DECOUPLING

With the PW basis $|\mathbf{k}, \mathbf{G}\rangle = e^{\mathrm{i}(\mathbf{k}+\mathbf{G})^\top \mathbf{r}}$, equation 35 can be decoupled into $|\mathcal{K}|$ independent equations where $\mathcal{K}$ is the k-path, since the Hamiltonian is block diagonal in $\mathbf{k}$. Firstly, the kinetic operator is diagonal in the PW basis. Since PW basis are orthogonal in both $\mathbf{G}$ and $\mathbf{k}$ index, i.e. $\langle\mathbf{k}', \mathbf{G}'|\mathbf{k}, \mathbf{G}\rangle = \delta_{\mathbf{k}\mathbf{k}'}\delta_{\mathbf{G}\mathbf{G}'}$, we have

$$T_{\mathbf{k}',\mathbf{G}';\mathbf{k},\mathbf{G}} = \langle\mathbf{k}', \mathbf{G}'|\hat{T}|\mathbf{k}, \mathbf{G}\rangle = \langle\mathbf{k}', \mathbf{G}'|\left[\frac{1}{2}\|\mathbf{k} + \mathbf{G}\|^2 |\mathbf{k}, \mathbf{G}\rangle\right] = \frac{1}{2}\|\mathbf{k} + \mathbf{G}\|^2 \delta_{\mathbf{k}\mathbf{k}'}\delta_{\mathbf{G}\mathbf{G}'}. \tag{36}$$

The potential operator is block diagonal in $\mathbf{k}$. Since $\hat{V}_{\text{eff}}$ is periodic, it can be expanded as $\hat{V}_{\text{eff}}(\mathbf{r}) = \sum_{\mathbf{G}} V_{\mathbf{G}}e^{\mathrm{i}\mathbf{G}^\top \mathbf{r}}$. Then

$$\langle\mathbf{k}', \mathbf{G}'|\hat{V}_{\text{eff}}|\mathbf{k}, \mathbf{G}\rangle = \sum_{\mathbf{G}''} V_{\mathbf{G}''}\frac{1}{\Omega_A}\int_{\Omega_A} \mathrm{d}\mathbf{r} \exp[\mathrm{i}(\mathbf{G}'' - \mathbf{k}' - \mathbf{G}' + \mathbf{k} + \mathbf{G})] \tag{37}$$

which must be zero for $\mathbf{k} \neq \mathbf{k}'$. Now, since the Hamiltonian is block diagonal in $\mathbf{k}$, its spectrum consists of the spectrum of each $\mathbf{k}$ block, and we can solve each $\mathbf{k}$ block separately.

This k-space decoupling actually holds for Hamiltonian basis $|\mathbf{k}, \alpha\rangle$ built from valid periodic basis $|\phi_\alpha\rangle$ for $u_{n\mathbf{k}}$ in general, and is not dependent on the basis used. Firstly, the potential operator $\hat{V}_{\text{eff}}$ is local, so it commutes with $e^{\mathrm{i}\mathbf{k}^\top \mathbf{r}}$, and therefore there are no coupling between different k. Secondly, the kinetic energy is invariant under a shift over the lattice vector $\mathbf{R}$, so we have

$$\begin{aligned} T_{\mathbf{k}'\alpha';\mathbf{k}\alpha} &= \frac{1}{N_k}\int_{N_k\Omega_A} \mathrm{d}\mathbf{r}\, \phi_{\mathbf{k}',\alpha'}(\mathbf{r})^*[-\frac{1}{2}\nabla^2]\phi_{\mathbf{k},\alpha}(\mathbf{r}) \\ T'_{\mathbf{k}'\alpha';\mathbf{k}\alpha} &= \frac{1}{N_k}\int_{N_k\Omega_A+\mathbf{R}} \mathrm{d}\mathbf{r}\, \phi_{\mathbf{k}',\alpha'}(\mathbf{r} - \mathbf{R})^*[-\frac{1}{2}\nabla^2]\phi_{\mathbf{k},\alpha}(\mathbf{r} - \mathbf{R}) \\ &= \frac{1}{N_k}\int_{N_k\Omega_A} \mathrm{d}\mathbf{r}\, e^{\mathrm{i}\mathbf{k}'^\top \mathbf{R}}\phi_{\mathbf{k}',\alpha'}(\mathbf{r})^*[-\frac{1}{2}\nabla^2]e^{-\mathrm{i}\mathbf{k}^\top \mathbf{R}}\phi_{\mathbf{k},\alpha}(\mathbf{r}) \\ &= e^{\mathrm{i}(\mathbf{k}'-\mathbf{k})^\top \mathbf{R}}T_{\mathbf{k}'\alpha';\mathbf{k}\alpha}. \end{aligned} \tag{38}$$

Now since $T = T'$ due to the translational invariance, $T_{\mathbf{k}'\alpha';\mathbf{k}\alpha}$ and $e^{i(\mathbf{k}'-\mathbf{k})^\top \mathbf{R}} \neq 0$ for $\mathbf{k} \neq \mathbf{k}'$, $T_{\mathbf{k}'\alpha';\mathbf{k}\alpha}$ must be zero for $\mathbf{k} \neq \mathbf{k}'$ which means that $\hat{T}$ is block diagonal in $\mathbf{k}$.

For each diagonal block of the Hamiltonian, define

$$\hat{H}_{\mathbf{k}}[\rho] := e^{-i\mathbf{k}\mathbf{r}}\hat{H}[\rho]e^{i\mathbf{k}\mathbf{r}} = e^{-i\mathbf{k}\mathbf{r}}(-\frac{1}{2}\nabla^2 + \hat{V}_{\text{eff}}[\rho])e^{i\mathbf{k}\mathbf{r}} = -\frac{1}{2}(i\mathbf{k} + \boldsymbol{\nabla})^2 + \hat{V}_{\text{eff}}[\rho] \quad (39)$$

where the product rule of Laplacian is used:

$$\nabla^2\left(e^{i\mathbf{k}^\top \mathbf{r}}f\right) = e^{i\mathbf{k}^\top \mathbf{r}}(\nabla^2 f - 2i\mathbf{k}\cdot\boldsymbol{\nabla}f - \|\mathbf{k}\|^2 f) \quad (40)$$

Now substitute the Bloch theorem 34 into the periodic KS equation (35), we get the following eigenvalue problem

$$\hat{H}_{\mathbf{k}}[\rho]u_{n\mathbf{k}} = \epsilon_{n\mathbf{k}}u_{n\mathbf{k}}, \quad (41)$$

and the Hamiltonian matrix is discretized as

$$H_{\mathbf{k}nm}[\rho] = \langle u_{n\mathbf{k}}|\hat{H}_{\mathbf{k}}[\rho]|u_{m\mathbf{k}}\rangle = \sum_{\alpha\beta} c^*_{n\mathbf{k}\alpha}c_{m\mathbf{k}\beta}H_{\mathbf{k},\alpha\beta}[\rho]. \quad (42)$$

## B  ORTHONORMALITY OF THE DPW BASIS

$$\langle\mathbf{G}|\mathbf{G}'\rangle = \frac{1}{\Omega}\int_{\Omega_A}\left(\big|\boldsymbol{\nabla}f^{-1}(\mathbf{r})\big|d\mathbf{r}\right)\exp\left[i(\mathbf{G}'-\mathbf{G})f^{-1}(\mathbf{r})\right]$$
$$= \frac{1}{\Omega}\int_\Omega d^3\xi\,\exp\left[i(\mathbf{G}'-\mathbf{G})\boldsymbol{\xi}\right] = \delta_{\mathbf{G}\mathbf{G}'}. \quad (43)$$

## C  RATIONAL-QUADRATIC SPLINE

Neural spline flow Durkan et al. (2019) creates a monotonic bijection $g$ on an 1D interval by dividing the interval to $K$ bins with $K+1$ knots $\{x^{(k)}\}_{k=0}^K$ where the $k$-th knot has height $y^{(k)}$ where $y^{(k)} < y^{(k+1)}$, and putting a rational-quadratic $g^{(k)}$ in each bin. The neighboring rational-quadratics connect smoothly at the knots with learnable slope $\delta^{(k)}$. $g^{(k)}$ interpolate the knots smoothly since $\frac{d}{dx}g^{(k)}(0) = \delta^{(k)}$ and $\frac{d}{dx}g^{(k)}(1) = \delta^{(k+1)}$. Furthermore, its inverse can be computed easily. The bijection constructed from these $K$ rational-quadratic functions is called the rational-quadratic spline (RQS).

For a given input $x$ in the $k$-th bin, denotes its relative coordinate within the bin as $\chi(x) = (x - x^{(k)})/w^{(k)}$, and the slope between the $k$-th and $k+1$-th knot as $s^{(k)} = (y^{(k+1)} - y^{(k)})/w^{(k)}$, where $w^{(k)} = x^{(k+1)} - x^{(k)}$ is the bin width, the rational-quadratic is defined as

$$f^{(k)}(\chi) = y^{(k)} + \frac{(y^{(k+1)} - y^{(k)})[s^{(k)}\chi^2 + \delta^{(k)}\chi(1-\chi)]}{s^{(k)} + [\delta^{(k+1)} + \delta^{(k)} - 2s^{(k)}]\chi(1-\chi)} \quad (44)$$

Next, let's derive this form from scratch. To create quadratic interpolation between knots $(x^{(k)}, y^{(k)})$ and $(x^{(k+1)}, y^{(k+1)})$, one can define

$$\alpha^{(k)}(\chi) = y^{(k+1)}\chi^2 + y^{(k)}(1-\chi)^2 \quad (45)$$

However, the gradient of the above function is given by

$$\frac{d}{dx}\alpha^{(k)}(\chi) = \frac{2}{w^{(k)}}[y^{(k+1)}\chi - y^{(k)}(1-\chi)]. \quad (46)$$

At knot points, the gradients are fixed to $2y^{(k+1)}/w^{(k)}$ and $-2y^{(k)}/w^{(k)}$. To be able to specify both the value and the first derivative at each knot point for the spline, Gregory and Delbourgo Gregory & Delbourgo (1982) proposed the following rational-quadratic $f^{(k)}$ where one could specify the first derivative at the knots $\delta^{(k)}, \delta^{(k+1)}$:

$$\alpha^{(k)}(\chi) = s^{(k)}[y^{(k+1)}\chi^2 + y^{(k)}(1-\chi)^2] + [y^{(k)}\delta^{(k+1)} + y^{(k+1)}\delta^{(k)}]\chi(1-\chi)$$
$$\beta^{(k)}(\chi) = s^{(k)}[\chi^2 + (1-\chi)^2] + [\delta^{(k+1)} + \delta^{(k)}]\chi(1-\chi) \quad (47)$$
$$f^{(k)}(\chi) = \frac{\alpha^{(k)}(\chi)}{\beta^{(k)}(\chi)}.$$

$f^{(k)}$ is still a quadratic interpolation since $f^{(k)}(0) = y^{(k)}$ and $f^{(k)}(1) = y^{(k+1)}$. Note that

$$
\begin{aligned}
\frac{\mathrm{d}}{\mathrm{d}x}\alpha^{(k)}(\chi) &= (x - x^{(k)})2[y^{(k+1)}\chi - y^{(k)}(1-\chi)] + [y^{(k)}\delta^{(k+1)} + y^{(k+1)}\delta^{(k)}](1-2\chi) \\
\frac{\mathrm{d}}{\mathrm{d}x}\beta^{(k)}(\chi) &= (x - x^{(k)})2[\chi - (1-\chi)] + [\delta^{(k+1)} + \delta^{(k)}](1-2\chi)
\end{aligned}
\tag{48}
$$

therefore

$$
\gamma^{(k)}(\chi) = \beta^{(k)}(\chi)\frac{\mathrm{d}}{\mathrm{d}x}\alpha^{(k)}(\chi) - \alpha^{(k)}(\chi)\frac{\mathrm{d}}{\mathrm{d}x}\beta^{(k)}(\chi) = (s^{(k)})^2[\delta^{(k+1)}\chi^2 + 2s^{(k)}\chi(1-\chi) + \delta^{(k)}(1-\chi)^2]
$$

$$
\frac{\mathrm{d}}{\mathrm{d}x}f^{(k)}(\chi) = \gamma^{(k)}(\chi)/[\beta^{(k)}(\chi)]^2.
\tag{49}
$$

It's easy to verify that $\frac{\mathrm{d}}{\mathrm{d}x}f^{(k)}(0) = \delta^{(k)}$ and $\frac{\mathrm{d}}{\mathrm{d}x}f^{(k)}(1) = \delta^{(k+1)}$. Inverting the function $f^{(k)}$ with given output $y$ amounts to solving the quadratic equation:

$$
a^{(k)}(\chi) - y\beta^{(k)}(\chi) = 0
\tag{50}
$$

whose solution is given by the quadratic formula

$$
x = \frac{1}{2a}[-b + \sqrt{b^2 - 4ac}]
\tag{51}
$$

where $a, b, c$ are the coefficients when the quadratic equation is written in the standard form of $ax^2 + bx + c = 0$.

## D    GRIDS

To generate grids used in computation, i.e., $\mathbf{r_n}, \mathbf{G_n}$ (see section A.2), we used the `fftfreq` function from JAX, which uses the **zero-first convention**, i.e., it always puts the zero element in the first position. Specifically, for both $\mathbf{r_n}$ and $\mathbf{G_n}$, the `fftfreq` function generates the $n_i$ values arranged as $[0, \ldots, \lfloor N_i/2 \rfloor, -\lfloor (N_i - 1)/2 \rfloor, \ldots, -1]$. In this work, we always use cubic grids, so $N_1 = N_2 = N_3 = N$, and the total grid size is $N^3$. Note that for density calculation, we need to double the grid size due to frequency doubling, so the total grid size is $(2N)^3$. We will note the $(2N)^3$ grid sizes explicitly to reduce the complexity of the main text.

## E    DIFFERENTIAL GEOMETRY

We will use Einstein notation where lower indices are covariant, upper indices are contravariant, and repeated indices are summed implicitly.

### E.1    CURVILINEAR COORDINATES OVER A RIEMANNIAN MANIFOLD

We work with two $n$-tori $M \simeq \mathbb{T}^n = \mathbb{R}^n/\mathbb{Z}^n$ and $N \simeq \mathbb{T}^n = \mathbb{R}^n/\mathbb{Z}^n$. Fix fundamental domains $U \subset M, V \subset N$ and coordinate charts $\xi : U \to \Omega, x : V \to \Omega, \Omega = [0,1]^n$, obtained from the quotient identification (periodic boundary conditions on $\partial\Omega$). Given a diffeomorphism $f : M \to N$, its coordinate representation is $\tilde{f} := x \circ f \circ \xi^{-1} : \Omega \longrightarrow \Omega$. We identify points on the manifold with their coordinates, and the diffeomorphism $f$ with its coordinate representation $\tilde{f}$, so we write $x = f(\xi)$.

$$
\begin{array}{ccc}
U \subset M & \xrightarrow{\quad f \quad} & V \subset N \\
{\scriptstyle \xi}\downarrow & & \downarrow{\scriptstyle x} \\
\Omega \subset \mathbb{R}^n & \xrightarrow{x \circ f \circ \xi^{-1}} & \Omega \subset \mathbb{R}^n
\end{array}
$$

Torus $N$ represents the physical space and coordinate $x$ is the usual Cartesian coordinate, while $M$ is a parameter torus where $\xi$ is a *curvilinear coordinate*.

### E.2 FIBER BUNDLE

A *smooth fiber bundle* $E$ over a manifold $M$ is a larger manifold that comes with a continuous surjection $\pi : E \to M$. $\pi$ attaches extra data from the fiber manifold $F$ to every point on the base manifold $M$, and $E$ is locally a product space: at each open neighbourhood $U \subset M$, the fiber is attached to the base manifold $M$ via the *local trivialization* $\phi_U : \pi^{-1}(U) \xrightarrow{\cong} U \times F$. On overlaps, trivializations are related by transition functions $g_{UV} : U \cap V \to G$ where the *structure group* $G$ can induce nontrivial topology. Since tori are oriented, most fiber bundles on them are *trivial*, i.e., $E \cong M \times F$ globally.

A *smooth section* $s : M \to E$ (for trivial bundle $s : M \to M \times F$) chooses an instance from the fiber $F$ for every point on the manifold $M$, and it satisfies $\pi \circ s = \mathrm{id}_M$. Intuitively, it is the inverse of $\pi$.

### E.3 PUSHFORWARDS AND PULLBACKS

For tori, the tangent bundle is a trivial vector bundle $TM \cong M \times \mathbb{R}^n$, and the fibers at point $x = f(\xi)$ are the tangent spaces $T_\xi M, T_x N$. Due to triviality, we have global bases (coordinate frames) for $TM$ and $TN$, which are the sets of partial derivative operators $\{ \frac{\partial}{\partial \xi^\alpha} \equiv \partial_\alpha \}$ and $\{ \frac{\partial}{\partial x^i} \equiv \partial_i \}$. To avoid ambiguity, the indices for $\xi$ are in Greek letters and indices for $x$ are in Roman letters. Any vector field on $N$ is a smooth section of $TN$, which can be expanded using the basis $\mathbf{v} = v^i \partial_i$, and similarly for $TM$.

At point $x = f(\xi)$, the pushforward $Tf : T_\xi M \to T_x N$ is the linearized $f$ which sends vectors from $T_\xi M$ to $T_x N$ using the chain rule:

$$Tf(\partial_\alpha) = \frac{\partial x^i}{\partial \xi^\alpha} \frac{\partial}{\partial x^i} = J_\alpha^i \partial_i. \tag{52}$$

So the component of $Tf(\partial_\alpha)$ is the $\alpha$-th column of the Jacobian $J_\alpha^i$. Since the Jacobian is invertible, we also have the pushforward of the inverse $Tf^{-1}(\partial_i) = (J^{-1})_i^\alpha \partial_\alpha$.

The cotangent bundles is also trivial for tori: $T^*M \cong M \times (\mathbb{R}^n)^*$, where $(\mathbb{R}^n)^*$ is the dual space of $\mathbb{R}^n$. The bases of the cotangent spaces $T_\xi^* M, T_r^* N$ are the linear functionals $\{ \mathrm{d}\xi^\alpha : T_\xi M \to \mathbb{R} \}$, $\{ \mathrm{d}x^i : T_r N \to \mathbb{R} \}$. It is defined as $\mathrm{d}x^i(\partial_j) = \delta_j^i, \mathrm{d}\xi^\alpha(\partial_\beta) = \delta_\beta^\alpha$, so for $\mathbf{v} = v^i \partial_i \in T_x N$ we have $\mathrm{d}x^i(\mathbf{v}) = v^i$. The cotangent basis is used to expand any covectors, for example, the differential of a scalar field $\phi$: $\mathrm{d}\phi = \partial_i \phi \, \mathrm{d}x^i$. The pullback $T^*f$ sends covectors from $T_x^* N$ to $T_\xi^* M$, again using the chain rule:

$$T^*f(\mathrm{d}x^i) = \mathrm{d}(x^i \circ f) = \frac{\partial x^i}{\partial \xi^\alpha} \mathrm{d}\xi^\alpha = J_\alpha^i \mathrm{d}\xi^\alpha. \tag{53}$$

To simplify notation, we will make pushforward and pullback implicit from now on:

$$\partial_\alpha = J_\alpha^i \partial_i, \quad \mathrm{d}x^i = J_\alpha^i \mathrm{d}\xi^\alpha. \tag{54}$$

### E.4 RIEMANNIAN METRIC

The Riemannian metric in the $x$ coordinate $g : T_x N \times T_x N \to \mathbb{R}$ is a $(0, 2)$-tensor that defines the inner product

$$g(\mathbf{v}, \mathbf{w}) := (g_{ij} \mathrm{d}x^i \otimes \mathrm{d}x^j)(\mathbf{v}, \mathbf{w}) = g_{ij} \mathrm{d}x^i(\mathbf{v}) \mathrm{d}x^j(\mathbf{w}), \tag{55}$$

where $\otimes$ is tensor product and $g_{ij} = g_{ji}$. We will use the standard inner product notation from now on, where $g$ is implicit: $\mathbf{v} \cdot \mathbf{w} = g(\mathbf{v}, \mathbf{w})$. The tensor component is given by $g_{ij} = \partial_i \cdot \partial_j$. The inverse metric has tensor component $g^{ij}$ defined as $g_{ik} g^{kj} = \delta_i^j$.

The metric tensor also allows us to raise and lower indices of tensor components, which is equivalent to mapping between the tangent bundle $TN$ and the cotangent bundle $T^*N$, through the musical isomorphism $\sharp$ and $\flat$: $\partial^i := (\mathrm{d}x^i)^\sharp$. For differential $\mathrm{d}\phi$, its Riesz representation is given by the (contravariant) gradient, $\mathrm{grad}\ \phi := (\mathrm{d}\phi)^\sharp = (\partial_i \phi \mathrm{d}x^i)^\sharp = \boldsymbol{\nabla}^i \phi\ \partial_i$, whose components is given by $\boldsymbol{\nabla}^i \phi = g^{ij} \partial_j \phi$. One can easily verify this: for any $\mathbf{v} \in T_x N$ we have $(\mathrm{grad}\ \phi) \cdot \mathbf{v} = g^{ij} \partial_j \phi \cdot \mathbf{v} = g^{ij} \mathrm{d}x^i(\partial_j \phi) \mathrm{d}x^j(\mathbf{v}) = \partial_i \phi \mathrm{d}x^i(\mathbf{v})$.

On the manifold $(N, x)$, we take the standard Euclidean metric $g_{ij} = \delta_{ij}$ since $N$ represents the physical space, and the n-tori $\mathbb{T}^n$ are flat. In curvilinear coordinate $\xi^\alpha$, the metric tensor component

can be computed using the pushforward

$$g_{\alpha\beta} = \partial_\alpha \cdot \partial_\beta = (J_\alpha^i \partial_i) \cdot (J_\beta^j \partial_j) = J_\alpha^i J_\beta^j \delta_{ij} = \mathbf{J}_\alpha \cdot \mathbf{J}_\beta. \tag{56}$$

Note that $g^{\alpha\beta} = (\mathbf{J}^{-1})^\alpha \cdot (\mathbf{J}^{-1})^\beta$, and $\partial^\alpha$ is the $\alpha$-th row of the inverse Jacobian:

$$\partial^\alpha = g^{\alpha\gamma}\partial_\gamma = [(\mathbf{J}^{-1})^\alpha \cdot (\mathbf{J}^{-1})^\gamma]\mathbf{J}_\gamma = (\mathbf{J}^{-1})^\alpha. \tag{57}$$

### E.5    COVARIANT DERIVATIVES

For scalar field $\phi$, we can get a covariant derivative from the gradient through musical isomorphism: $\mathbf{\nabla}\phi = (\text{grad } \phi)^\flat$. To generalize the covariant derivative to vector fields, we also need to consider the infinitesimal change in the coordinate frame. Hence $\mathbf{\nabla}$ is uniquely defined through the *connection*, i.e. its action on the tangent space basis: $\mathbf{\nabla}_i \partial_j = \Gamma_{ij}^k \partial_k$, where $\Gamma_{ij}^k$ is called the Christoffel symbols. Given $\Gamma_{ij}^k$, the component of the covariant derivative for any vector field $\mathbf{v} = v^i \partial_i$ is given by

$$(\mathbf{\nabla}_i v)^k = \partial_i v^k + \Gamma_{ij}^k v^j. \tag{58}$$

which can be thought of as applying the chain rule to the product of a vector component and the basis.

We can also take the covariant derivative along a general direction $\mathbf{w}$: $\mathbf{\nabla}_\mathbf{w} = w^i \mathbf{\nabla}_i$. Furthermore, $\mathbf{\nabla}$ provides a way to measure parallel lines in curve space: along a $C^1$ curve $\gamma(t)$, a vector field $\mathbf{v}$ is parallel if and only if it satisfies the ODE which says the variation of vector components $v^k$ in the direction of tangent $\dot\gamma$ is zero:

$$\mathbf{\nabla}_{\dot\gamma} v^k(t) = \mathbf{\nabla}_{\dot r^i \partial_i} v^k(t) = \dot r^i (\partial_i v^k + \Gamma_{ij}^k v^j) = 0. \tag{59}$$

Without extra constraints, $\Gamma_{ij}^k$ can take an arbitrary value, so there are many possible connections. However, the fundamental theorem of Riemannian geometry states that there is a unique affine connection called the *Levi-Civita connection* that is *torsion-free*:

$$T(\partial_i, \partial_j) = \mathbf{\nabla}_i \partial_j - \mathbf{\nabla}_j \partial_i - [\partial_i, \partial_j] = 0, \tag{60}$$

and *metric-compatible*: $\mathbf{\nabla} g = 0$. For coordinate frames, the Lie bracket vanish $[\partial_i, \partial_j] = 0$ so the torsion-free condition implies the symmetry

$$\mathbf{\nabla}_i \partial_j = \mathbf{\nabla}_j \partial_i \quad \Rightarrow \quad \Gamma_{ij}^k = \Gamma_{ji}^k. \tag{61}$$

For the metric compatibility condition, expanding the tensor component of $\mathbf{\nabla} g$ with the Leibniz rule gives

$$0 = \mathbf{\nabla}_k g_{ij} = \partial_k g_{ij} - g(\mathbf{\nabla}_k \partial_i, \partial_j) - g(\partial_i, \mathbf{\nabla}_k \partial_j) = \partial_k g_{ij} - \Gamma_{ki}^\ell g_{\ell j} - \Gamma_{kj}^\ell g_{i\ell}. \tag{62}$$

Since $g_{\ell j}$ lowers the indices we can write $\Gamma_{jki} := \Gamma_{ki}^\ell g_{\ell j}$. Cyclically permuting the indices yields two more equalities

$$\partial_j g_{ki} = \Gamma_{ijk} + \Gamma_{kji}, \quad \partial_i g_{jk} = \Gamma_{kij} + \Gamma_{jik}. \tag{63}$$

Combine the three equalities from metric compatibility and use the symmetry $\Gamma_{ij}^k = \Gamma_{ji}^k$ to yield a formula that computes the Levi-Civita connection $\Gamma_{ij}^k$ from the metric tensor $g_{ij}$:

$$\Gamma_{kj}^\ell = \frac{1}{2} g^{\ell i} (\partial_k g_{ij} + \partial_i g_{jk} - \partial_j g_{ik}). \tag{64}$$

In $x$ coordinate, $g_{ij}$ is the constant $\delta_{ij}$ so $\partial_k g_{ij} = 0$, and therefore $\Gamma_{ij}^k = 0$ everywhere, which means the covariant derivatives are just the normal gradient. But in $\xi$ coordinate $\Gamma_{\alpha\beta}^\gamma$ is non-trivial since $g_{\alpha\beta}$ is given by the Jacobian of the diffeomorphism $f$, which is parameterized by a normalizing flow. From now on, we refer to the Levi-Civita connection as simply the connection.

### E.6    DIVERGENCE AND THE LAPLACE-BELTRAMI OPERATOR

Just like the connection $\Gamma_{ij}^k$ defines the covariant derivative $\mathbf{\nabla}$, the *contracted connection* $A_\beta := \Gamma_{\alpha\beta}^\alpha$ defines the divergence operator $\mathbf{\nabla}\cdot$ in $\xi$ coordinate using the definition (Eq. 58):

$$\mathbf{\nabla} \cdot \mathbf{v} = (\mathbf{\nabla}_\alpha v)^\alpha = \partial_\alpha v^\alpha + \Gamma_{\alpha\beta}^\alpha v^\beta = (\partial_\alpha + A_\alpha) v^\alpha. \tag{65}$$

Contracting eq. 62 with $g^{ij}$ yields

$$g^{ij}\partial_k g_{ij} = \Gamma^\ell_{ki}\delta^i_\ell + \Gamma^\ell_{kj}\delta^j_\ell = 2\Gamma^\ell_{k\ell} = 2A_k. \tag{66}$$

Therefore $A_\beta = \frac{1}{2}g^{\alpha\gamma}\partial_\beta g_{\alpha\gamma}$. Note that the determinants of the metric tensor and the Jacobians are related as $|g|^{\frac{1}{2}} = |J|$. For invertible matrix $M$, Jacobi formula states that $\partial_\beta|M| = |M|\operatorname{tr}(M^{-1}\partial_\beta M)$, so we have $\partial_\beta|J| = \partial_\beta|g|^{\frac{1}{2}} = \frac{1}{2}|g|^{-\frac{1}{2}}\operatorname{tr}(g^{-1}\partial_\beta g) = \frac{1}{2}|J|g^{\alpha\gamma}\partial_\beta g_{\gamma\alpha}$, and:

$$\partial_\beta \log|J| = |J|^{-1}\partial_\beta|J| = \frac{1}{2}g^{\alpha\gamma}\partial_\beta g_{\gamma\alpha} = A_\beta. \tag{67}$$

We see that $A_\beta$ is also the differential of the log determinant of the Jacobian (LDJ), which can be easily computed when the diffeomorphism $f$ is a normalizing flow.

The Laplace-Beltrami operator $\Delta$ is the divergence of the gradient. For scalar field $\phi$ we have

$$\begin{aligned}
\Delta\phi = \boldsymbol{\nabla}\cdot(\operatorname{grad}\phi) &= (\partial_\alpha + A_\alpha)(g^{\alpha\beta}\partial_\beta\phi)\\
&= (\partial_\alpha g^{\alpha\beta})\partial_\beta\phi + g^{\alpha\beta}\partial_\alpha\partial_\beta\phi + (|J|^{-1}\partial_\alpha|J|)g^{\alpha\beta}\partial_\beta\phi\\
&= |J|^{-1}[|J|(\partial_\alpha g^{\alpha\beta})\partial_\beta\phi + |J|g^{\alpha\beta}(\partial_\alpha\partial_\beta\phi) + g^{\alpha\beta}\partial_\beta\phi\,(\partial_\alpha|J|)]\\
&= |J|^{-1}\partial_\alpha\left(|J|g^{\alpha\beta}\partial_\beta\phi\right).
\end{aligned} \tag{68}$$

### E.7 DIFFERENTIAL FORMS AND INTEGRATION

Differential $n$-forms are multilinear functionals that measure signed $n$-dimensional volume, which are the integrands on a manifold. Any $n$-form $\omega$ can be represented with the basis $\mathrm{d}^n\xi$ where

$$\omega = \omega_{1\ldots n}\mathrm{d}^n\xi, \qquad \mathrm{d}^n\xi := \bigwedge_{\alpha=1}^n \mathrm{d}\xi^\alpha = \sum_{\sigma\in P(n)}\operatorname{sgn}(\sigma)\bigotimes_{\alpha=1}^n \mathrm{d}\xi^\alpha. \tag{69}$$

Here $P$ is the permutation group over $\{1,\ldots,n\}$, and $\wedge$ is the exterior product, which is the antisymmetrized tensor product. For example, the 2-form basis $\mathrm{d}^2\xi = \mathrm{d}\xi^1\otimes\mathrm{d}\xi^2 - \mathrm{d}\xi^2\otimes\mathrm{d}\xi^1$ measures the signed area. We see that any 2-form component $\omega_{ij}$ must be antisymmetric as well, i.e. $\omega_{ij} = -\omega_{ji}$, and this is true for $n$-form component in general. The collection of $n$-form spaces over the manifold $M$ is $\wedge^n(T^*M) = \{\omega_{1\ldots n}\wedge_{\alpha=1}^n\mathrm{d}\xi^\alpha \mid \xi\in M\}$, $n$-form bundle, which is a real line bundle. From the definition, it is easy to see that the pullback rule of a 1-form (eq. 53) generalizes to $n$-form as

$$T^*f(\mathrm{d}^n x) = |J|\mathrm{d}^n\xi, \quad T^*f^{-1}(\mathrm{d}^n\xi) = |J|^{-1}\mathrm{d}^n x. \tag{70}$$

Integration on manifolds is defined as pullback to Euclidean space $\mathbb{R}^n$

$$\int_M \omega := \int_U T^*x(\omega). \tag{71}$$

where $x: U\subset\mathbb{R}^n\to M$ is a coordinate chart. In our case, coordinate $x$ is identified with the points on the physical tori $N$, so for a top form $\rho\,\mathrm{d}^n x$, we have

$$\int_\Omega \rho(x)\,\mathrm{d}^n x = \int_{f^{-1}(\Omega)} T^*f(\rho(x)\,\mathrm{d}^n x) = \int_{f^{-1}(\Omega)} |J|\rho(\xi)\mathrm{d}^n\xi, \tag{72}$$

and similarly

$$\int_\Omega \rho(\xi)\,\mathrm{d}^n\xi = \int_{f(\Omega)} T^*f^{-1}(\rho(\xi)\,\mathrm{d}^n\xi) = \int_{f(\Omega)} |J|^{-1}\rho(x)\mathrm{d}^n x, \tag{73}$$

where $J^i_\alpha = \frac{\partial x^i}{\partial\xi^\alpha}$, which is the usual change of variable formula.

### E.8 WEIGHTED IBP

Suppose $u, w$ are scalar fields and $\mathbf{v}$ is a vector field, and $u, w, \mathbf{v}$ are periodic over the cell $\Omega$. We can define a weighted divergence as $w^{-1}\partial_\alpha(wv^\alpha)$. By the product rule

$$\partial_\alpha(wuv^\alpha) = w(\partial_\alpha u)v^\alpha + u\partial_\alpha(wv^\alpha) = w\left[(\partial_\alpha u)v^\alpha + u\,w^{-1}\partial_\alpha(wv)\right]. \tag{74}$$

Integrate over $\Omega$, use divergence theorem and the fact that $uw\mathbf{v}$ is periodic we get

$$\int_\Omega \partial_\alpha(wuv^\alpha)\mathrm{d}^3\xi = \int_{\partial\Omega} wuv^\alpha\mathbf{n}\,\mathrm{d}S = 0. \tag{75}$$

So we have the following weighted integration by parts (IBP) identity

$$\int_\Omega (\partial_\alpha u)v^\alpha\mathrm{d}^3\xi = -\int_\Omega uw^{-1}\partial_\alpha(wv^\alpha)\mathrm{d}^3\xi. \tag{76}$$

### E.9 DENSITY, HALF-DENSITY AND SCALAR BUNDLE

For probability densities, events are measured in $L^1$. For wavefunctions describing bound states, events are measured in $L^2$ instead. These normalization constraints need to be *invariant* under diffeomorphism. Since densities are like volume, we define the *density bundle* as the unsigned $n$-form bundle $|\wedge^n(T^*M)|$, where the pullback is the unsigned version of Eq. 72

$$T^*f(\mathrm{d}^n x) = |J||\mathrm{d}^n \xi|, \quad T^*f^{-1}(|\mathrm{d}^n \xi|) = |J|^{-1}|\mathrm{d}^n x|, \tag{77}$$

which ensures invariance of volume under diffeomorphism. And naturally, wavefunctions live in the *half-density bundle* $|\wedge^n(T^*M)|^{\frac{1}{2}}$ which is a *complex* line bundle with basis $|\mathrm{d}^n \xi|^{\frac{1}{2}}$ that is the square root of unsigned top form. This encodes the $L^2$ integrability of the half-densities. The pullback is exactly the square root of Eq. 77

$$T^*f(|\mathrm{d}^n x|^{\frac{1}{2}}) = |J|^{\frac{1}{2}}|\mathrm{d}^n \xi|^{\frac{1}{2}}, \quad T^*f^{-1}(|\mathrm{d}^n \xi|^{\frac{1}{2}}) = |J|^{-\frac{1}{2}}|\mathrm{d}^n x|^{\frac{1}{2}}. \tag{78}$$

which ensures the invariance of normalization half-densities under change of coordinate: for $\Phi, \Psi \in |\wedge^n(T^*M)|^{\frac{1}{2}}$, $\Phi^*\Phi \in |\wedge^n(T^*M)|$ and we have $\int \Phi^*\Phi \mathrm{d}^n x = \int \Phi^*\Phi|J||\mathrm{d}^n \xi|$. Since we never do computation on the k-form bundle in this paper, we will omit the absolute sign in the integral in this paper.

Next, we derive the bilinear form with $\Delta$ on the half-density bundle. Given half-densities $\Psi, \Phi \in |\wedge^n(T^*M)|^{\frac{1}{2}}$, using weighted IBP (Eq. 76) with $w = |J|$, and the identity $A_\beta = |J|^{-1}\partial_\beta|J|$ for contracted connection (Eq. 67), we have

$$\int_\Omega \partial_\alpha(|J|uv^\alpha)d^3\xi = \int_\Omega [(\partial_\alpha|J|)uv^\alpha + |J|\partial_\alpha(uv^\alpha)]\mathrm{d}^3\xi = 0, \tag{79}$$

and therefore

$$\int_\Omega (|J|^{-\frac{1}{2}}\Phi^*)\Delta_{\boldsymbol{x}}(|J|^{-\frac{1}{2}}\Psi)\mathrm{d}^3 x$$

$$= \int_\Omega \Phi^*|J|^{-1}\partial_\alpha(|J|g^{\alpha\beta}\partial_\beta\Psi)\ \mathrm{d}^3\xi \tag{80}$$

$$= -\int_\Omega [(-\tfrac{1}{2}A_\alpha + \partial_\alpha)\Phi^*]g^{\alpha\beta}[(-\tfrac{1}{2}A_\beta + \partial_\beta)\Psi]\ \mathrm{d}^3\xi.$$

To remove the factor $\frac{1}{2}$, we defined the recaled contracted connection $A' = \frac{1}{2}A$.

## F DETAILS ON THE METRIC-WEIGHTED DENSITY MATRIX

Denote the occupation vector at $\mathbf{k}$ as $\mathbf{f_k} \in \mathbb{R}^{N_{\text{band}}}$, $\mathbf{F_k} = \mathrm{diag}(\mathbf{f_k})$ and the band-resolved density matrix as

$$\Gamma_{\mathbf{k},nm}(\mathbf{r}) = u_{n\mathbf{k}}^*(\mathbf{r})u_{m\mathbf{k}}(\mathbf{r}), \quad \rho(\mathbf{r}) = \sum_{\mathbf{k}} \mathrm{tr}[\mathbf{F_k}\Gamma_{\mathbf{k},nm}(\mathbf{r})]. \tag{81}$$

Recall that in the FDPW basis

$$u_{n\mathbf{k}}(\mathbf{r}) = \frac{1}{\sqrt{\Omega}}|J|^{-\frac{1}{2}}\sum_{\mathbf{G}} c_{n\mathbf{k}\mathbf{G}}e^{i\mathbf{G}^\top f^{-1}(\mathbf{r})}, \tag{82}$$

and its on the distorted grid $\{\mathbf{r}_i = f(\boldsymbol{\xi}_i)\}_{i=1}^N$ can be computed via FFT:

$$\mathbf{u}_{n\mathbf{k}} = \frac{N}{\sqrt{\Omega}}\mathbf{J}^{-\frac{1}{2}}\mathrm{FFT}^{-1}(\mathbf{c}_{n\mathbf{k}}) \in \mathbb{C}^{N_{\text{basis}}}. \tag{83}$$

Then the evaluation of $\Gamma_{\mathbf{k},nm}(\mathbf{r})$ on the distorted grid can be calculated as

$$\boldsymbol{\Gamma}_{\mathbf{k},nm} = \mathbf{u}_{n\mathbf{k}}^* \odot \mathbf{u}_{m\mathbf{k}} = \frac{1}{\Omega}\mathbf{J}^{-1}[N^2\mathrm{FFT}^{-1}(\mathbf{c}_{n\mathbf{k}})^* \odot \mathrm{FFT}^{-1}(\mathbf{c}_{m\mathbf{k}})]. \tag{84}$$

Similarly, we define $S_{\mathbf{k},nm} = S_{n\mathbf{k}}^*(\mathbf{r})S_{m\mathbf{k}}(\mathbf{r})$ and $S(\mathbf{r}) = \sum_{\mathbf{k}} \mathrm{tr}[\mathbf{F_k}S_{\mathbf{k},nm}(\mathbf{r})] = |J|\rho(\mathbf{r})$. Under FDPW basis, for any operator $\hat{O}$, the matrix element $O_{\mathbf{k},\mathbf{G}'\mathbf{G}} := \langle \mathbf{G}' + \mathbf{k}|\hat{O}|\mathbf{G} + \mathbf{k}\rangle$ has the pullback

$$O_{\mathbf{k},\mathbf{G}'\mathbf{G}} = \frac{1}{\Omega_A}\int_{\Omega_A} |J|^{-1}e^{-i(\mathbf{G}'+\mathbf{k})\cdot f^{-1}(\mathbf{r})}\hat{O}e^{i(\mathbf{G}+\mathbf{k})\cdot f^{-1}(\mathbf{r})}\mathrm{d}^3 r$$

$$= \frac{1}{\Omega}\int_\Omega e^{-i(\mathbf{G}'+\mathbf{k})^\top \boldsymbol{\xi}}[T^*f(\hat{O})]e^{i(\mathbf{G}+\mathbf{k})^\top \boldsymbol{\xi}}\mathrm{d}^3\xi, \tag{85}$$

and we have $\langle\psi_{n\mathbf{k}}|\hat{O}|\psi_{m\mathbf{k}}\rangle = \sum_{\mathbf{G}'\mathbf{G}} c^*_{n\mathbf{k}\mathbf{G}'} c_{m\mathbf{k}\mathbf{G}} O_{\mathbf{k},\mathbf{G}'\mathbf{G}}$. For local operator $O(\mathbf{r})$, the pullback is simply function composition $T^* f(O) = O \circ f$ and $O_{\mathbf{k},\mathbf{G}'\mathbf{G}} = \frac{1}{\Omega}\int_\Omega e^{i(\mathbf{G}-\mathbf{G}')^\top \boldsymbol{\xi}} O(f(\boldsymbol{\xi}))\mathrm{d}^3\xi$ which is independent of $\mathbf{k}$. Furthermore it is *diagonal*: $O_{\mathbf{G}'\mathbf{G}} = \delta_{\mathbf{G}'\mathbf{G}} O_{\mathbf{0}\mathbf{G}}$. Let $\mathbf{O}$ be the evaluation of $O(\mathbf{r})$ on the distorted grid $\{\mathbf{r}_i = f(\boldsymbol{\xi}_i)\}_{i=1}^N$, we have (see Appendix N on the prefactor)

$$
\begin{aligned}
\langle\psi_{n\mathbf{k}}|\hat{O}|\psi_{m\mathbf{k}}\rangle &= \sum_{\mathbf{G}'\mathbf{G}} \int_\Omega [c_{n\mathbf{k}\mathbf{G}'}\phi_{\mathbf{G}'}(\boldsymbol{\xi})]^* O(\boldsymbol{\xi})[c_{m\mathbf{k}\mathbf{G}}\phi_{\mathbf{G}}(\boldsymbol{\xi})]\mathrm{d}^3\xi \\
&\simeq \sum_{\mathbf{G}'\mathbf{G}} \frac{\Omega}{N} \sum_{i=1}^N [c_{n\mathbf{k}\mathbf{G}'}\phi_{\mathbf{G}'}(\boldsymbol{\xi})]^* O(\boldsymbol{\xi})[c_{m\mathbf{k}\mathbf{G}}\phi_{\mathbf{G}}(\boldsymbol{\xi})] \\
&= \frac{\Omega}{N} \left[\frac{N^2}{\Omega} \mathrm{FFT}^{-1}(\mathbf{c}_{n\mathbf{k}})^* \odot \mathrm{FFT}^{-1}(\mathbf{c}_{m\mathbf{k}})\right]^\dagger \mathbf{O} \\
&= \frac{\Omega}{N} \mathbf{S}^\dagger_{\mathbf{k},nm} \mathbf{O}.
\end{aligned}
\tag{86}
$$

## G  Laplacian-Beltrami operator under the FDPW basis

Recall that DPW are regular plane waves in the parameter space $\phi_{\mathbf{G}}(\boldsymbol{\xi}) = \frac{1}{\sqrt{\Omega}}\exp(i\mathbf{G}^\top\boldsymbol{\xi})$. Let $\Psi = \phi_{\mathbf{G}}$ and $\Phi = \phi_{\mathbf{G}'}$, then using bilinear form of $\Delta$ on the half-density bundle (Eq. 80), we can compute the matrix element of $\Delta$ under the DPW basis.

$$
\begin{aligned}
&\langle\psi_{n\mathbf{k}}|\hat{T}|\psi_{m\mathbf{k}}\rangle \\
&= \sum_{\mathbf{G}'\mathbf{G}} c^*_{n\mathbf{k}\mathbf{G}'} c_{m\mathbf{k}\mathbf{G}} \langle\mathbf{G}'+\mathbf{k}|-\frac{1}{2}\Delta|\mathbf{G}+\mathbf{k}\rangle \\
&= \frac{1}{2} \sum_{\mathbf{G}'\mathbf{G}} \int_\Omega [c^*_{n\mathbf{k}\mathbf{G}'}(-A'_\alpha + \partial_\alpha)\phi^*_{\mathbf{G}'+\mathbf{k}}] g^{\alpha\beta} [c_{m\mathbf{k}\mathbf{G}}(-A'_\beta + \partial_\beta)\phi_{\mathbf{G}+\mathbf{k}}]\mathrm{d}^3\xi \\
&\approx \frac{1}{2} \sum_{\mathbf{G}'\mathbf{G}} \frac{\Omega}{N} \sum_{i=1}^N [c^*_{n\mathbf{k}\mathbf{G}'}(-A'_\alpha(\boldsymbol{\xi}_i) + \partial_\alpha)\phi^*_{\mathbf{G}'+\mathbf{k}}(\boldsymbol{\xi}_i)] g^{\alpha\beta}(\boldsymbol{\xi}_i) [c_{m\mathbf{k}\mathbf{G}}(-A'_\beta(\boldsymbol{\xi}_i) + \partial_\beta)\phi_{\mathbf{G}+\mathbf{k}}(\boldsymbol{\xi}_i)] \\
&= \frac{1}{2}\frac{\Omega}{N} \sum_{i=1}^N \mathbf{W}^*_{\alpha,n\mathbf{k}}(\boldsymbol{\xi}_i) g^{\alpha\beta}(\boldsymbol{\xi}_i) \mathbf{W}_{\beta,m\mathbf{k}}(\boldsymbol{\xi}_i),
\end{aligned}
\tag{87}
$$

where the summation is over uniform $\boldsymbol{\xi}$-space grid and

$$
\mathbf{W}_{\beta,n\mathbf{k}}(\boldsymbol{\xi}) = \frac{N}{\sqrt{\Omega}}[-A'_\beta(\boldsymbol{\xi})\mathrm{FFT}^{-1}(\mathbf{c}_{n\mathbf{k}}) + \mathrm{FFT}^{-1}(i(\mathbf{G}+\mathbf{k})\mathbf{c}_{n\mathbf{k}})] \in \mathbb{C}^{N_{\text{basis}}},
\tag{88}
$$

where we used the fact that $\langle\phi_{\mathbf{G}}|\partial_\alpha\psi_{n\mathbf{k}}\rangle = i(\mathbf{G}+\mathbf{k})c_{n\mathbf{k}\mathbf{G}}$.

## H  Spectral form of the potential operators

We first introduce some shorthands similar to Rostgaard (2009). Given $\rho_1, \rho_2 : \Omega_A \to \mathbb{R}$ that are periodic and zero-mean over $\Omega_A$, the interaction energy between them under a shift-invariant kernel $K(r)$:

$$
\begin{aligned}
(\rho_1|K(r)|\rho_2) &:= \langle\rho_1 \star K|\rho_2\rangle \\
&= \int_{\Omega_A} \left(\int_{\mathbb{R}^3} K(\|\mathbf{r}-\mathbf{r}'\|)\rho_1(\mathbf{r}')\mathrm{d}\mathbf{r}'\right) \rho_2(\mathbf{r})\mathrm{d}\mathbf{r},
\end{aligned}
\tag{89}
$$

and for Coulomb interaction:

$$
(\rho_1|\rho_2) := (\rho_1|\frac{1}{r}|\rho_2) \quad ((\rho_1)) := (\rho_1|\rho_1).
\tag{90}
$$

We further define the potential generated from $\rho_1$ as $V_{\rho_1}$ so that $(\rho_1|\rho_2) = \langle\rho_1 \star \frac{1}{r}|\rho_2\rangle = \langle V_{\rho_1}|\rho_2\rangle$. Denote the atomic point charge as $\rho^{\text{atom}}(\mathbf{r}) = -\sum_\ell Z_\ell \delta(\mathbf{r}-\boldsymbol{\tau}_\ell)$ and the electronic density as $\rho$, the total potential energy is

$$
\frac{1}{2}((\rho + \rho^{\text{atom}})) = \frac{1}{2}((\rho)) + \frac{1}{2}((\rho^{\text{atom}})) + (\rho|\rho^{\text{atom}})
\tag{91}
$$

where the $\frac{1}{2}$ prefactor prevents double counting.

The potential energy is only conditionally convergent as both $((\rho))$ and $(\rho|\rho^{\text{atom}})$ diverge. This can be shown by some Fourier analysis. The Coulombic potential generated from a charge distribution $\rho_0$ has a simple diagonal representation in the frequency space

$$
\begin{aligned}
\tilde{V}(\mathbf{G}) &= \lim_{\alpha \to 0} \mathcal{F}\left[-\frac{1}{4\pi}\nu_\alpha \star -4\pi\rho_0\right](\mathbf{G}) \\
&= \lim_{\alpha \to 0} \tilde{\nu}_\alpha(\mathbf{G})\tilde{\rho}_0(\mathbf{G}) = \lim_{\alpha \to 0} \frac{4\pi\tilde{\rho}_0(\mathbf{G})}{\|\mathbf{G}\|^2 + \alpha^2}.
\end{aligned}
\tag{92}
$$

where $\mathcal{F}$ is the Fourier transform operator, and $\nu_\alpha(r) = \frac{e^{-\alpha r}}{r}$ is the Yukawa kernel (see Appendix I). The energy $\langle V|\rho_1\rangle = \sum_{\mathbf{G}} \tilde{V}(\mathbf{G})\tilde{\rho}_1(\mathbf{G})$ is clearly divergent due to the singularity of $\tilde{V}(\mathbf{G})$ at $\mathbf{G} = \mathbf{0}$.

On the other hand, the singularity at $\mathbf{G} = \mathbf{0}$ can be avoided by using a charge neutral $\rho$, i.e. $\tilde{\rho}(\mathbf{G}) = \mathbf{0}$. And one can neutralize any charge distribution with total charge $Z_{\text{tot}}$ by adding an uniform background charges $\rho^\pm(\mathbf{r}) = \mp Z_{\text{tot}}/\Omega$. Thus, we can define convergent potential energies as

$$
\frac{1}{2}((\rho + \rho^{\text{atom}})) = \underbrace{\frac{1}{2}((\rho + \rho^+))}_{\text{Hartree}} + \underbrace{(\rho + \rho^+|\rho^{\text{atom}} + \rho^-)}_{\text{External}} + \underbrace{\frac{1}{2}((\rho^{\text{atom}} + \rho^-))}_{\text{Nucleus}}.
\tag{93}
$$

The reciprocal representation of the Hartree and the external potential can then be obtained through equation 92 by setting $\rho_0$ to $\rho + \rho^+$ and $\rho^{\text{atom}} + \rho^-$ respectively.

With the FDPW basis, one can no longer perform basis projection by doing FFT over a uniform grid in $\Omega$. DPW basis does become regular PW in the parameter space $\Omega$, and as mentioned in section G, the matrix elements of the kinetic operator can be evaluated using FFT with a uniform grid in $\Omega$. However, the Yukawa kernel in the parameter space $\nu_\alpha \circ f$ is not longer spherical symmetric due to the distortion $f$, so one can no longer obtain a simple expression of its projection to $e^{i\mathbf{G}^\top \boldsymbol{\xi}}$ by using spherical coordinate in $\Omega$ (see Appendix I).

## I  Yukawa kernel

The electrostatic Poisson equation can be solved in the Fourier space:

$$
\nabla^2 V(\mathbf{r}) = -4\pi\rho(\mathbf{r}) \Rightarrow -\|\mathbf{G}\|^2 \tilde{V}(\mathbf{G}) = -4\pi\tilde{\rho}(\mathbf{G}) \Rightarrow \tilde{V}(\mathbf{G}) = \frac{4\pi}{\|\mathbf{G}\|^2}\tilde{\rho}(\mathbf{G}).
\tag{94}
$$

Note that at $\mathbf{G} = \mathbf{0}$ we have a singularity, so this ill-defined unless $\tilde{\rho}(\mathbf{0}) = 0$. Therefore the Fourier transform of the Coulomb potentials $V(\mathbf{r}) = \frac{1}{r}$ can only be defined via a limit:

$$
\tilde{V}(\mathbf{G}) = \lim_{\alpha \to 0} \mathcal{F}\left[-\frac{1}{4\pi}\nu_\alpha \star -4\pi\rho\right](\mathbf{G}) = \lim_{\alpha \to 0} \tilde{\nu}_\alpha(\mathbf{G})\tilde{\rho}(\mathbf{G}),
\tag{95}
$$

where $\nu_\alpha(r) = \frac{e^{-\alpha r}}{r}$ is the Yukawa kernel whose Fourier transform can be calculated using spherical coordinates

$$
\begin{aligned}
\tilde{\nu}^\alpha(\mathbf{G}) &= \int_{\mathbb{R}^3} d\mathbf{r}\, \nu_\alpha(\mathbf{r})e^{-i\mathbf{G}\cdot\mathbf{r}} \\
&= \int_0^{2\pi}\int_0^\pi\int_0^\infty \frac{e^{-\alpha r}}{r}e^{-i\|\mathbf{G}\|r\cos\theta}r^2\sin\theta\, drd\theta d\phi \\
&= \frac{4\pi}{\|\mathbf{G}\|^2 + \alpha^2}.
\end{aligned}
\tag{96}
$$

Therefore

$$
\tilde{V}(\mathbf{G}) = \lim_{\alpha \to 0} \frac{4\pi\tilde{\rho}(\mathbf{G})}{\|\mathbf{G}\|^2 + \alpha^2}.
\tag{97}
$$

## J   THE CHARGE NEUTRALITY REQUIREMENT

$$\int_{\Omega_A} \mathrm{d}\mathbf{r}\, V(\mathbf{r}) = 0. \tag{98}$$

With a charge neutral $\rho$, we have

$$\tilde{\rho}(\mathbf{0}) = \int_{\Omega_A} \mathrm{d}\mathbf{r}\, e^{-\mathrm{i}\mathbf{0}\cdot\mathbf{r}}\rho(\mathbf{r}) = 0, \tag{99}$$

and from Appendix I, the singularity at $\mathbf{G} = \mathbf{0}$ is removed:

$$\tilde{V}(\mathbf{0}) = \lim_{\alpha \to 0} \frac{4\pi\tilde{\rho}(\mathbf{0})}{\alpha^2} = \lim_{\alpha \to 0} \frac{0}{\alpha^2} = 0. \tag{100}$$

Alternatively, one can derive the charge neutrality requirement without doing any Fourier analysis as well. For periodic $\rho$, $V_H$ is also periodic:

$$V_H(\mathbf{r} + \mathbf{R}) = \sum_{\mathbf{n}} \int_{\Omega + \mathbf{R}_{\mathbf{n}}} \frac{1}{\|\mathbf{r} + \mathbf{R} - \mathbf{r}'\|}\rho(\mathbf{r}')\mathrm{d}\mathbf{r}' = \sum_{\mathbf{n}} \int_{\Omega - \mathbf{R} + \mathbf{R}_{\mathbf{n}}} \frac{1}{\|\mathbf{r} - \mathbf{r}'\|}n(\mathbf{r}')\mathrm{d}\mathbf{r}' = V_H(\mathbf{r}). \tag{101}$$

Therefore, we need to impose periodic boundary conditions on the unit cell $\Omega_A$ when solving the Poisson equation for a periodic density $\rho$. This introduces the constraint of *charge neutrality* for $\rho$. This is because

$$\int_{\Omega_A} \mathrm{d}\mathbf{r}\, \rho(\mathbf{r}) = \int_{\Omega_A} \mathrm{d}\mathbf{r}\, \nabla^2 V(\mathbf{r}) = \oint_{\partial\Omega_A} \mathrm{d}\mathbf{S}\, \nabla V(\mathbf{r}) = 0. \tag{102}$$

The second equality is due to the divergence theorem. The final equality holds since the integral contributions from opposite edges of $\Omega_A$ cancel out, as $\nabla V(\mathbf{r})$ are the same due to PBC and $\mathrm{d}\mathbf{S}$ has the opposite sign.

## K   POISSON SUMMATION

For an arbitrary function $f$, the Fourier coefficient of its periodization over a lattice (also shifted by $\boldsymbol{\tau}$) is given by:

$$\begin{aligned}
\frac{1}{\Omega_A} \int_{\Omega_A} \mathrm{d}\mathbf{r} \left[\sum_{\mathbf{R}} f(\mathbf{r} - \boldsymbol{\tau} - \mathbf{R})\right] e^{-\mathrm{i}\mathbf{G}\cdot\mathbf{r}} &= \frac{1}{\Omega_A} \sum_{\mathbf{R}} \int_{\Omega_A} \mathrm{d}\mathbf{r}\, f(\mathbf{r} - \boldsymbol{\tau} - \mathbf{R})e^{-\mathrm{i}\mathbf{G}\cdot(\mathbf{r}-\mathbf{R})} \quad (e^{\mathrm{i}\mathbf{G}\cdot\mathbf{R}} = 1)\\
&= \frac{1}{\Omega_A} \sum_{\mathbf{R}} \int_{\Omega_A + \mathbf{R}} \mathrm{d}\mathbf{r}'\, f(\mathbf{r}' - \boldsymbol{\tau})e^{-\mathrm{i}\mathbf{G}\cdot\mathbf{r}'} \quad (\mathbf{r}' = \mathbf{r} - \mathbf{R})\\
&= \frac{1}{\Omega_A} \int_{\mathbb{R}^3} \mathrm{d}\mathbf{r}'\, f(\mathbf{r}' - \boldsymbol{\tau})e^{-\mathrm{i}\mathbf{G}\cdot\mathbf{r}'}\\
&= \frac{1}{\Omega_A} e^{-\mathrm{i}\mathbf{G}\cdot\boldsymbol{\tau}} \int_{\mathbb{R}^3} \mathrm{d}\mathbf{r}''\, f(\mathbf{r}'')e^{-\mathrm{i}\mathbf{G}\cdot\mathbf{r}''} \quad (\mathbf{r}'' = \mathbf{r}' - \boldsymbol{\tau})\\
&= \frac{1}{\Omega_A} \tilde{f}(\mathbf{G})e^{-\mathrm{i}\mathbf{G}\cdot\boldsymbol{\tau}}
\end{aligned} \tag{103}$$

where $\tilde{f}(\mathbf{G})$ is the (continuous) Fourier transform of $f$, and $\Omega_A$ is the unit cell of the lattice. The gaining of phase factor $e^{-\mathrm{i}\mathbf{G}\cdot\boldsymbol{\tau}}$ is also known as the shift theorem. We can now represent the periodization as a Fourier series:

$$\sum_{\mathbf{R}} f(\mathbf{r} - \boldsymbol{\tau} - \mathbf{R}) = \frac{1}{\Omega_A} \sum_{\mathbf{G}} \tilde{f}(\mathbf{G})e^{-\mathrm{i}\mathbf{G}\cdot\boldsymbol{\tau}}. \tag{104}$$

This is known as the Poisson summation.

## L  EXTERNAL POTENTIAL

The detailed derivation for the Ewald summation of the external potential:

$$
V_{\text{ext}}(\mathbf{r})
$$

$$
= \sum_{\mathbf{G} \neq \mathbf{0}} \left[ \sum_{\ell} \tilde{V}(\mathbf{G}; Z_\ell) e^{-i\mathbf{G}\cdot\boldsymbol{\tau}_\ell} \right] e^{i\mathbf{G}\cdot\mathbf{r}}
$$

$$
= \sum_{\ell} \left\{ \sum_{\mathbf{G}} [\tilde{V} - \tilde{V}_\eta](\mathbf{G}; Z_\ell) e^{i\mathbf{G}\cdot(\mathbf{r}-\boldsymbol{\tau}_\ell)} + \sum_{\mathbf{G} \neq \mathbf{0}} \tilde{V}_\eta(\mathbf{G}; Z_\ell) e^{i\mathbf{G}\cdot(\mathbf{r}-\boldsymbol{\tau}_\ell)} - [\tilde{V} - \tilde{V}_\eta](\mathbf{0}; Z_\ell) \right\} \quad (105)
$$

$$
= \sum_{\ell} \left\{ \sum_{\mathbf{R}} [V - V_\eta](\mathbf{r} - \boldsymbol{\tau}_\ell - \mathbf{R}; Z_\ell) + \sum_{\mathbf{G} \neq \mathbf{0}} \tilde{V}_\eta(\mathbf{G}; Z_\ell) e^{i\mathbf{G}\cdot(\mathbf{r}-\boldsymbol{\tau}_\ell)} - [\tilde{V} - \tilde{V}_\eta](\mathbf{0}; Z_\ell) \right\}
$$

where in the last equality we used Poisson summation (see Appendix K). Note that the reciprocal vectors $\mathbf{G}$ in the above equation are on $\Omega$ instead of $\Omega$ as in the other expression involving DPW. In periodic systems, the last term $[\tilde{V} - \tilde{V}_\eta](\mathbf{0}; Z_\ell)$ can be dropped since we use charge-neutral density.

The real space summation will decay rapidly since both function decays into $1/r$. However, the Coulomb potential $V(r; Z) = \frac{Z}{r}$ has a singularity near $r = 0$, so real space summation would require very high resolution around the origin. Specifically, because the bare nuclear Coulomb potential $\frac{1}{r}$ is non-analytic at the origin, the exact orbitals possess a Kato cusp and the associated fields are not smooth on the computational torus. Spectral/PW discretizations deliver exponential/spectral convergence only for analytic targets; a cusp instead forces the Fourier/spectral coefficients to decay only algebraically, which in turn makes total energies approach the CBS limit at a polynomial rate as the basis is refined. In this work, we use Analytical Norm-Conserving (ANC) regularized potential Gygi (2023), which is a spherical local all-electron pseudopotential given by

$$
V_{\text{ANC}}(r; 1) = -\frac{1}{2} + \frac{1}{r} h'(r) + \frac{1}{2} h''(r) + \frac{1}{2} h'(r)^2
$$
$$
h(r; a, b) = -r \operatorname{erf}(ar) + b e^{-a^2 r^2} \quad (106)
$$
$$
V_{\text{ANC}}(r; Z) = Z^2 V_{\text{ANC}}(Zr; 1).
$$

where $Z$ is the charge of the associated atom nucleus. There are two parameters $a, b$, but the $b$ parameter is determined by $a$ through the norm-conservation constraint, and this mapping from $a$ to $b$ is precomputed and tabulated. The ANC potential is identical to Coulomb outside a small core but smooth at the origin, and is analytic, which means spectral convergence is possible.

And as discussed in Lindsey & Sharma (2024), although both $\tilde{V}_{\text{ANC}}(\mathbf{0}; 1) = \int_0^\infty \mathrm{d}r\, r^2 V_{\text{ANC}}(r)$ and $\tilde{V}_\eta(\mathbf{0}; 1) = \int_0^\infty \mathrm{d}r\, r^2 V_\eta(r)$ diverges, but the difference is bounded. So this term can be numerically calculated by selecting a radial cutoff $R$ where the difference between the two functions becomes very small:

$$
\tilde{V}_{\text{ANC}}(\mathbf{0}; 1) \approx \int_0^R \mathrm{d}r\, r^2 V_{\text{ANC}}(r; 1), \qquad\qquad \tilde{V}_{\text{ANC}}(\mathbf{0}; Z) \approx \frac{1}{Z} \int_0^{RZ} \mathrm{d}r\, r^2 V_{\text{ANC}}(r; 1)
$$

$$
\tilde{V}_\eta(\mathbf{0}; 1) \approx \int_0^R \mathrm{d}r\, r^2 V_\eta(r; 1), \qquad\qquad \tilde{V}_\eta(\mathbf{0}; Z) = Z \tilde{V}_\eta(\mathbf{0}; 1)
$$

$$
= -\frac{1}{4} \left( \frac{2R}{\eta\sqrt{\pi} e^{\eta^2 R^2}} + \left( -\frac{1}{\eta^2} + 2R^2 \right) \operatorname{erf}(\eta R) \right)
$$

$$(107)$$

An approximation formula for $\tilde{V}_{\text{ANC}}(\mathbf{0}; 1)$ can be found in Lindsey & Sharma (2024).

## M  EWALD SUMMATION OF THE NUCLEUS POTENTIAL ENERGY

The Ewald summation of the nucleus potential energy is given by

$$
E_{\text{nuc}} = E_{\text{nuc}}^{\text{sr}} + E_{\text{nuc}}^{\text{lr}} - E_{\text{nuc}}^{\text{self}} \quad (108)
$$

where the short-range part $E_{\text{nuc}}^{\text{sr}}$ is given by the real space summation over the Ewald simulation cell $L$

$$E_{\text{nuc}}^{\text{sr}} \approx \frac{1}{2} \sum_{\ell} \sum_{\ell'} \sum_{\mathbf{n}}^{L} Z_\ell Z_{\ell'} \frac{\text{erfc}(\eta \|\boldsymbol{\tau}_\ell - \boldsymbol{\tau}_{\ell'} - \mathbf{R_n}\|)}{\|\boldsymbol{\tau}_\ell - \boldsymbol{\tau}_{\ell'} - \mathbf{R_n}\|} - \frac{\pi Z_{\text{tot}}^2}{2\Omega \eta^2}, \tag{109}$$

The long-range part $E_{\text{nuc}}^{\text{lr}}$ is given by the reciprocal space summation over the Ewald reciprocal lattice $L'$

$$E_{\text{nuc}}^{\text{lr}} \approx \frac{2\pi}{\Omega_A} \sum_{\mathbf{G} \neq \mathbf{0}}^{L'} \frac{1}{\|\mathbf{G}\|^2} \exp\left(-\frac{\|\mathbf{G}\|^2}{4\eta^2}\right) \left[\sum_\ell Z_\ell e^{i\mathbf{G} \cdot \boldsymbol{\tau}_\ell}\right]^2, \tag{110}$$

and the self-interaction correction in the long-range part is

$$E_{\text{nuc}}^{\text{self}} = \sum_\ell Z_\ell^2 \eta / \sqrt{\pi}. \tag{111}$$

## N  INTEGRATION FACTORS

For real space integration with FFT mesh, we multiply by the volume factor $\Omega_A/N$ where $N$ is the mesh size

$$\int_{\Omega_A} \mathrm{d}\mathbf{r}\, f(\mathbf{r})g(\mathbf{r}) \approx \frac{\Omega_A}{N} \sum_{i=1}^{N} f_i g_i. \tag{112}$$

## O  FREE-SPACE HARTREE GAUGE FOR FINITE SYSTEMS

For non-periodic systems embedded in a large cubic box $\Omega_A = [-\frac{a}{2}, \frac{a}{2}]^3$, the periodic Poisson solve on the torus fixes the gauge by setting the DC Fourier mode to zero, $\tilde{V}_H(\mathbf{G} = \mathbf{0}) = 0$. This is convenient but inconsistent with the free-space reference $V_H(\mathbf{r}) \to 0$ as $\|\mathbf{r}\| \to \infty$. We therefore align the gauge *post hoc* on the converged fields, without re-solving Poisson.

We emulate free space with the truncated Coulomb Green's function

$$g_{R_c}(\mathbf{r}) = \frac{\mathbf{1}[\|\mathbf{r}\| < R_c]}{\|\mathbf{r}\|}, \qquad \lim_{\mathbf{G} \to 0} \tilde{g}_{R_c}(\mathbf{G}) = 2\pi R_c^2. \tag{113}$$

Replacing the DC mode by $\tilde{V}_H(\mathbf{0}) \leftarrow 2\pi R_c^2 \bar{\rho}$ adds a constant shift

$$c_{\text{fs}} = 2\pi R_c^2 \bar{\rho}, \qquad \bar{\rho} = \frac{N_e}{\Omega_A}, \tag{114}$$

where $N_e$ is the total electron count. With our discrete Fourier convention (inverse FFT multiplies $1/N$, where $N$ is the total mesh size), the stored DC entry equals $N c_{\text{fs}}$.

The required quantities are computed on the parameter-space grid $\Omega = [-\pi, \pi]^3$ using the Jacobian $J = \frac{\partial r^i}{\partial \xi^\alpha}$:

$$N_e = \int_{\Omega_A} \rho(\mathbf{r})\, \mathrm{d}\mathbf{r} = \int_\Omega |J(\boldsymbol{\xi})|\, \rho(f(\boldsymbol{\xi}))\, \mathrm{d}\boldsymbol{\xi} \approx \sum_{i=1}^{N} |J(\boldsymbol{\xi}_i)|\, \rho_i \frac{\Omega}{N}, \quad \Omega = (2\pi)^3. \tag{115}$$

We select $R_c$ from the cell geometry. Let $\{\mathbf{a}_i\}_{i=1}^3$ be the cell vectors and $L_i = \|\mathbf{a}_i\|$. A robust default is $R_c = \frac{1}{2} \min_i L_i$, with an optional relaxation toward the molecular radius as in the implementation.

Finally, the Hartree energy is corrected analytically:

$$E_H^{\text{fs}} = E_H^{\text{per}} + \tfrac{1}{2} c_{\text{fs}} N_e, \tag{116}$$

which follows from $E_H = \frac{1}{2} \int_{\Omega_A} \rho\, V_H$ when $V_H$ is shifted by a constant. We report $E_{\text{total}}^{\text{fs}} = E_{\text{total}}^{\text{per}} + \frac{1}{2} c_{\text{fs}} N_e$, and retain $\tilde{V}_H(\mathbf{0})$ for completeness.

Figure 3 shows the $H_2$ dissociation curve computed with FDPW under this free-space alignment (left) alongside a Gaussian-orbital (PySCF/LDA) reference (right).

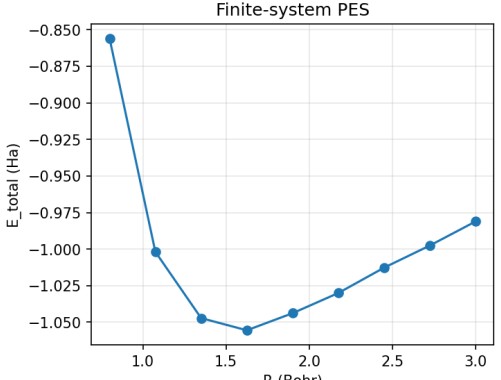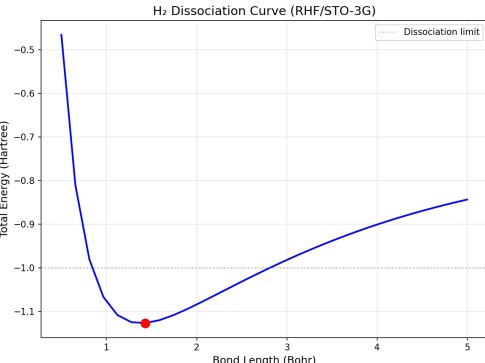

Figure 3: $H_2$ dissociation curves: FDPW finite-system (left, free-space Hartree gauge alignment) and Gaussian-orbital (PySCF/LDA) reference (right). Energies are total electronic energies (Ha) versus bond length $R$ (Bohr).

Table 2: Essential hyperparameters

| Parameter | Value |
|---|---|
| Autoregressive layers | 4 |
| Conditioner layers | 2 |
| Hidden size | 64 |
| Bins | 7 |
| Fourier features | 2 |
| Base range | [-3.1415926536, 3.1415926536] |
| Min bin size | 0.001 |
| Min knot slope | 0.001 |
| Max slope | 100.0 |
| Ground-state epochs | 3000 |
| Ground-state learning rate | 0.01 |
| Ground-state weight decay | 0.0 |
| Density-fit epochs | 3000 |
| Density-fit learning rate | 0.0002 |
| Shear regularization $\mu_{\text{shear}}$ | 0.005 |
| Trace regularization $\mu_{\text{smooth}}$ | 0.005 |

## P HYPERPARAMETERS

All essential hyperparameter are in Table 2.

## Q EXPERIMENTS ON FINITE SYSTEMS

We validate FDPW on the CO molecule (LDA_X) in a cubic box. PySCF was computed with DFT+LDA_X with cc-pVQZ basis. Empirically we find that all energy terms of FDPW the besides $E_{\text{ext}}$ (due to the use of ANC pseudopotential) reaches the PySCF reference energy within 5Ha at $N = 18\text{-}24$, whereas PW is nowhere near convergence.

Distorted grid and density at Figure 4.

## R AMORTIZATION OF THE PCG SOLVER

Solver traces in Figure 5 show stable Hartree energy and Poisson residual decay across epochs for the two runs; both reach the same fixed point in $E_H$.

Table 3: CO (LDA_X) finite-system energies (Ha) by method and grid size. Nucleus replusion $E_{nn} = 22.5352$ is same for all runs.

| Method | $N$ | $E_{\text{tot}}$ | $E_{\text{kin}}$ | $E_{\text{H}}$ | $E_{\text{ext}}$ | $E_{\text{x/xc}}$ |
|---|---|---|---|---|---|---|
| PySCF (ref) | - | $-111.5265$ | $111.5553$ | $75.9626$ | $-309.6100$ | $-11.9696$ |
| PW | 18 | $-112.1016$ | $56.0008$ | $65.1025$ | $-246.7359$ | $-9.0042$ |
| | 24 | $-184.0885$ | $90.3419$ | $57.8731$ | $-344.2963$ | $-10.5425$ |
| | 30 | $-94.6457$ | $66.2604$ | $69.9866$ | $-243.1868$ | $-10.2412$ |
| FDPW | 18 | $-116.3511$ | $105.9456$ | $73.8686$ | $-305.8903$ | $-12.8102$ |
| | 24 | $-119.0776$ | $114.1526$ | $74.1201$ | $-316.9114$ | $-12.9741$ |

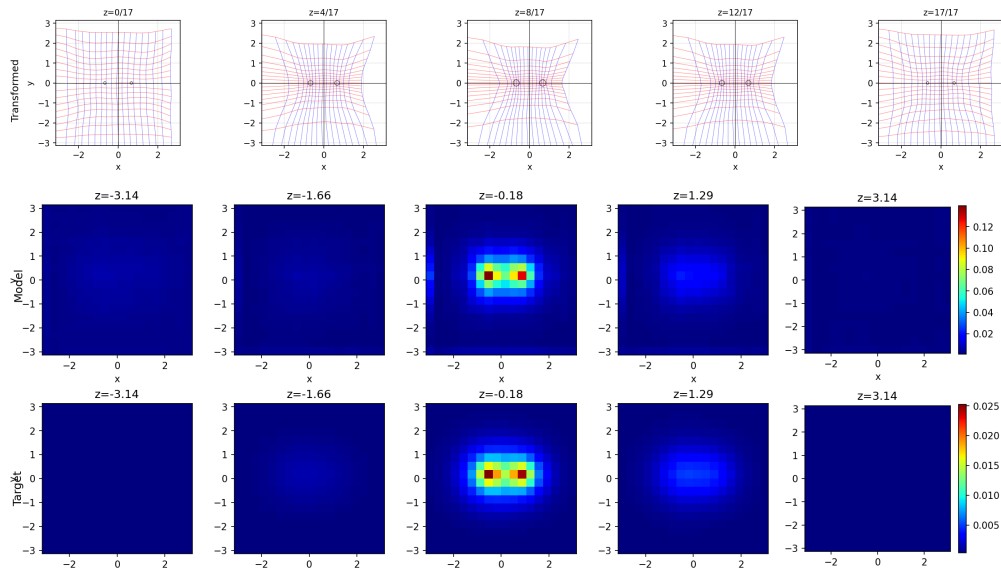

Figure 4: CO finite system: distorted grid (top) and resulting electron density slices (bottom). The grid is generated by the fitted flow using the prescribed density initialization; density shown on the distorted grid highlights resolution near nuclei and along the bond.

## S  PROFILING OF PROJECTOR-AUGEMNTED WAVE (PAW)

We profiled GPAW Mortensen et al. (2024) which is a well-established PW+PAW implementation. We used the direct optimization solver in GPAW to provide a fair comparison with our FDPW approach. We analyze the scaling of the PAW method over the diamond crystal with increasing plane-wave cutoff. The result is shown in Figure 6

We can observe the following:

1. As cutoff increase, the calculation of the XC energy over 3D grid via FFT dominates the computational cost and scales slightly better than $N \log N$

2. At small cutoff, time consumption of the atomic XC correction is comparable to that of the 3D grid XC calculation. Since the atomic XC correction scales with the highest angular momentum of the projectors and the size of the radial grid of the projector as well as the size of the spherical integration grid but not on the size of the G grid.

Therefore, if the plane wave cutoff is much larger compared to the atomic number, evaluating the XC energy of the 3D grid dominates the computation and scales as FFT. However, in practical calculation, there is evidence showing that the XC correction is the most time consuming part of the PAW computation. Intuitively this calculation is of order $O(N_{sphere} \times N_{radial} \times L)$ where $N_{sphere}, N_{radial}$ are the size of the spherical and radial grid for atomic calculation and $L = (2l+1)^2$

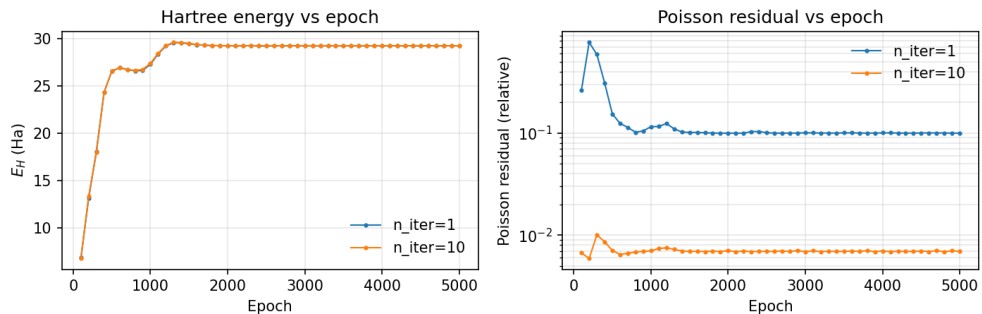

Figure 5: Hartree energy $E_H$ (left) and Poisson residual (right) vs. epoch for two finite-system runs with different $n_{\text{iter}}$. Residuals decay smoothly (log scale), and $E_H$ stabilizes as the fixed point is approached.

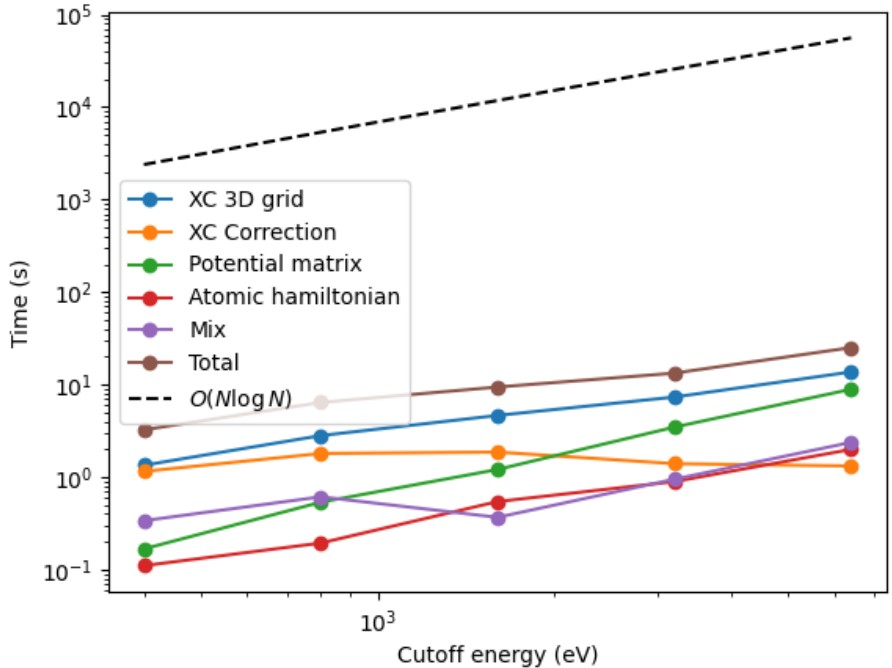

Figure 6: Hartree energy $E_H$ (left) and Poisson residual (right) vs. epoch for two finite-system runs with different $n_{\text{iter}}$. Residuals decay smoothly (log scale), and $E_H$ stabilizes as the fixed point is approached.

where $l$ is the highest angular momentum of the projectors. Since our method does not include this atomic calculation, the scalability of our method and PAW may not be comparable.

## T    USAGE OF LLM

LLM was used to this work to polish the writing of the paper.

