# OpenReview forum: "Flow-Distorted Plane Waves"
_ICLR.cc/2026/Conference — Submitted to ICLR 2026_

### Official Review · Reviewer_A7Dq · 2025-10-27

**Soundness:** 2
**Presentation:** 2
**Contribution:** 3
**Rating:** 4
**Confidence:** 2

**Summary:**

The paper proposes Flow-Distorted Plane Waves (FDPW): a geometry-aware basis obtained by composing a lattice linear map $T$ with a learned torus diffeomorphism $g_\theta$ while retaining PW-style algebra (FFT/NUFFT and k-point decoupling via a modified Bloch phase). The only "learning" component is a one-shot KL fitting of $g_\theta$ to a prescribed density, after which $g_\theta$ is frozen and the electronic structure problem is solved on the distorted basis.

**Strengths:**

- **Algebraic elegance and clarity.**
  Recasting the Bloch phase as $ \exp(i\,k^\top f^{-1}(r)) $ is a neat device that simultaneously addresses the $k$–space coupling due to $T$ and the nonlinear phase distortion due to $g$, yielding block-diagonal structure across $k$ after the warp.
- **Computationally coherent PW replacement.**
  Local operators are evaluated via two inverse FFTs and pointwise products; the kinetic energy becomes a “minimal‑coupling” quadratic form over the warped metric, avoiding dense $G{\times}G'$ matrices while preserving PW‑like efficiency.
- **Compact warp parameterization.**
  A circular RQS autoregressive flow on the torus offers sufficient expressivity with a small parameter count, suggesting a practical path to spatial adaptivity without abandoning spectral tooling.
- **Potential for efficiency gains.**
  The premise—achieving target accuracy with fewer modes/smaller grids by concentrating resolution near physically singular/rapid‑variation regions—addresses a real bottleneck in PW‑style solvers. Flag For Ethics Review

No ethics review needed

**Weaknesses:**

(W1) Empirical evidence

-	The present experimental scope appears limited (e.g., one crystalline material and one small molecule).
-	A quantitative, matched-accuracy comparison with recent adaptive bases/coordinate approaches, as well as strong PW+PAW/USPP setups, would help readers gauge relative merits in time/memory/accuracy.
-	Ablations seem absent for:
(i) NUFFT parameters (oversampling, kernel width) versus accuracy and wall-clock;
(ii) flow capacity/regularization;
(iii) the contribution of the prescribed-density fit versus simpler prescriptions/no fit;
(iv) Hartree/Poisson preconditioner robustness.
A compact table of matched-accuracy results and a few ablation curves would substantiate the efficiency claims more convincingly.

(W2) Learning is used as a pre-fit; $g_\theta$ is then fixed.
The ML component is currently a one-shot geometry pre-fit,
\\[\min_{\theta}\ \mathrm{KL}\\big(p_\theta \,\|\, \rho_{\mathrm{prescribed}}\big),\qquad p_\theta \text{ induced by } g_\theta,\\]
after which $g_\theta$ becomes a fixed, differentiable background for operator evaluation. In essence this is a basis change (a warped PW basis).

-	Why is learning necessary at all? Would an analytic/hand-crafted warp (e.g., radial crowding near nuclei plus lattice symmetries) achieve similar results? Please quantify.
-	Why not adapt $g_\theta$ during SCF/geometry optimization with stability safeguards (slow updates, penalties, bilevel schedules)? If attempted, please report stability and net benefit; otherwise the ML contribution may be viewed as just implementation convenience rather than learning.

(W3) Writing/positioning for an ML venue

The abstract, introduction, and related work are too terse for orienting ML readers: the problem, background, and significance are not sufficiently articulated, and the method exposition mixes core ideas with plumbing. It may help to separate “concept $\to$ algebraic consequences $\to$ implementation modules” and to add a brief “method-at-a-glance” figure.

**Questions:**

- **(Q1)** Clarify roles of $T$ vs $g$ in Eq. (6). Even with $g=\mathrm{Id}$, the lattice map $T$ generally couples $(G,k)$, breaking cross-$k$ orthogonality unless the Bloch phase is expressed through $f^{-1}$. The manuscript should explicitly separate: (a) the linear mismatch introduced by $T$ (which is induced by problems), and (b) the nonlinear distortion from $g$ (which is induced by methods).
- **(Q2)** On the representation $g=\mathrm{Id}+g_p$ and its scope. Arguments that absorb the periodic part into the cell-periodic function rely on $g$ being an orientation-preserving torus diffeomorphism homotopic to identity (as realized by the circular-flow family). They do not cover flips such as $g=-\mathrm{Id}$. Please state this assumption explicitly and indicate which steps (e.g., triangular Jacobian/log-det structure) rely on it.
- **(Q3)** There are several formatting issues.
  - Lines 248 and 249 are too close.
  - The format of the symbol "-" seems incorrect, e.g., in line 149.
  - There is a typo in lines 90–91: “when we G in subscript we mean…”.

---

> ### Author Response · Authors · 2025-11-15
>
> We thank the reviewer for the insightful and meticulous review. Here we address each of the weaknesses and questions you raised.
>
> (W1)
> > The present experimental scope appears limited
>
> Please see our common reply (R1).
>
> > A quantitative, matched-accuracy comparison with recent adaptive bases/coordinate approaches, as well as strong PW+PAW/USPP setups, would help readers gauge relative merits in time/memory/accuracy.
>
> As explained in (R2), USPP/PAW is orthogonal to our method and can actually be used together. (R3) covers the comparison with other adaptive bases methods.
>
> > Ablations seem absent for: (i) NUFFT parameters (oversampling, kernel width) versus accuracy and wall-clock; (ii) flow capacity/regularization; (iii) the contribution of the prescribed-density fit versus simpler prescriptions/no fit; (iv) Hartree/Poisson preconditioner robustness. A compact table of matched-accuracy results and a few ablation curves would substantiate the efficiency claims more convincingly.
>
> i. NUFFT: Please see (R7).
> ii. We acknowledge this point. Please see (R8).
> iii. As explained in (R6), a simpler fit, like radial crowding you suggested, would not work in our case. No-fit is equivalent to a regular planewave, which we already compared in Table 1.
> iv. The Hartree/Poisson preconditioner we used is simple, and there are no hyperparameters. We are not sure how it can be ineffective. Please advise on potential experiments to conduct on this, if you have any idea.
>
> (W2)
> > Would an analytic/hand-crafted warp achieve similar results?
>
> Please see (R6).
>
> We would also like to clarify that although we employ normalizing flow, currently the sole purpose of the flow is to create an optimizable bijection on 3-tori. In a sense, our method is more optimization than learning since the method is data-free, akin to PINN.
>
> > The ML component is currently a one-shot geometry pre-fit
>
> We acknowledge this point. One of the intentions of introducing the normalizing flow is to enable future data-driven training, as explained in (R2). We plan to do this in the future, but it remains out of the scope of this paper.
>
> > Why not adapt $g_\theta$ during SCF/geometry optimization with stability safeguards (slow updates, penalties, bilevel schedules)? If attempted, please report stability and net benefit; otherwise the ML contribution may be viewed as just implementation convenience rather than learning.
>
> It has already been discussed in the prior literature that adapting the basis during SCF is, in general, not a good idea, due to the contention between basis adaptation and energy minimization. A much simpler scheme would be to further fine-tune the basis after converging to the ground state with the initial adaptation. We did not include this as there is enough content in the current paper already, and this step does not add much value to the story we try to convey. As for why flow is used here, and possible future development, see (R6).
>
> (W3)
> Please see (R4).
>
> (Q1)
> It is true that when g=Id, the lattice map T will still break cross-k orthogonality, but this is trivial, since T can be absorbed into G and k, and the basis reduces to regular PW. Specifically $e^{ik^\top f^{-1}(r)}=e^{i k_A^\top r}$, which is just the standard Bloch phase factor. We will revise the writing of the Bloch phase factor section to make it clearer.
>
> (Q2)
> Thanks for your sharp observation. Indeed, our flow does not cover the case of $g=-Id$, which is still a valid bijection, but such a choice does not reduce the cutoff, since it is just a linear transformation. We will add this point to the final version of the paper.
>
> (Q3)
> Thanks for your detailed review. We will fix these formatting issues, along with other typos, in the final version of the paper.

---

### Official Review · Reviewer_MiZ7 · 2025-10-29

**Soundness:** 3
**Presentation:** 1
**Contribution:** 2
**Rating:** 2
**Confidence:** 3

**Summary:**

The paper introduces Flow‑Distorted Plane Waves (FDPW), a Galerkin basis that composes a periodic, bijective normalizing flow on the 3‑torus with plane‑wave coordinates to obtain adaptive spatial resolution while retaining much of the plane‑wave algebra (FFT/NUFFT structure). The authors provide a rigorous mathematical formulation grounded in differential geometry, treating wavefunctions as half-densities. A key theoretical contribution is the introduction of a modified Bloch phase factor, which elegantly preserves k-space decoupling in periodic systems—a necessity for efficient solid-state calculations.

However, this coordinate transformation introduces significant complexity. The Laplacian becomes the Laplace-Beltrami operator, requiring the evaluation of complex geometric quantities (Jacobian, metric tensor, connections) derived from the flow. Crucially, the Hartree potential can no longer be solved directly in Fourier space and requires an iterative Preconditioned Conjugate Gradient (PCG) solver. External potentials are handled via Non-Uniform FFTs (NUFFT).

The authors demonstrate the method on simple systems (Diamond, CO, H2), suggesting that FDPW can achieve comparable accuracy to standard PWs with a smaller basis set size (N). While this suggests potential memory savings, the analysis of the introduced computational overheads and the comparison against standard practices in the field are insufficient to establish practical superiority.

**Strengths:**

The paper presents a mathematically sophisticated approach to adaptive basis sets, with notable theoretical innovations.

1. Elegant Solution to K-Space Decoupling: The introduction of the modified Bloch phase (Sec 4.2) to maintain orthogonality between different k-points in a distorted coordinate system is a significant theoretical contribution. It elegantly resolves a fundamental barrier to applying DPW methods in periodic systems.

2. Rigorous Geometric Formulation: The authors provide a thorough derivation based on differential geometry (Appendices E, F, G). The treatment of wavefunctions as half-densities ensures the unitarity of the transformation and provides a sound basis for deriving the operators.

3. Novel Synthesis: Combining normalizing flows, a modern ML technique, with the established DPW framework offers a parameter-efficient and differentiable approach to defining the adaptive basis.

**Weaknesses:**

While the approach is theoretically strong, the paper has significant limitations regarding the scope of validation, analysis of computational overhead, and practical applicability.

1. Insufficient Comparisons to Standard Practices: The comparison is limited to vanilla Plane Waves. FDPW is presented as an all-electron method (using ANC potentials). To assess practical relevance, comparisons against highly optimized PW codes using standard pseudopotentials (e.g., PAW or ultrasoft), which inherently reduce the required basis size, are necessary. Comparisons with other modern adaptive basis methods (e.g., Lindsey & Sharma (2024)) are also missing.

2. Dependence on Heuristic Initialization: The method's performance relies on a two-stage process where the flow is first fitted to a heuristic "prescribed density" (Sec 4.4), involving several hyperparameters (Eq 13) and regularization weights (Eq 15). The paper does not sufficiently explore the robustness and sensitivity of the results to this initialization procedure.

3. Unanalyzed and Potentially Prohibitive Computational Overhead: FDPW introduces substantial overhead per iteration. This includes evaluating the flow and computing complex geometric quantities via automatic differentiation at every grid point. The kinetic energy evaluation (Eq. 18) and the iterative nature of the PCG solver also add significant cost. The paper lacks a breakdown of these costs and their scaling, making the true efficiency trade-off unclear.

4. Numerical Stability and Conditioning: There is a fundamental tension between adaptation (requiring strong distortions) and numerical stability. Strong distortions lead to ill-conditioned metric tensors, which can slow down the PCG solver and increase quadrature errors. This trade-off is not analyzed.

5. Limited Empirical Validation and Missing Features: The experiments are restricted to very simple systems (Diamond, H2, CO). Performance on realistic scenarios (metals, defects) remains unexplored. Furthermore, the framework lacks support for essential features like non-local pseudopotentials and efficient force calculations.

6. Accessibility: The paper heavily relies on advanced concepts from differential geometry (e.g., fiber bundles, pullbacks, connections) and solid-state physics. While rigorous, this presentation style makes the paper highly inaccessible to a general ICLR audience, with crucial details buried in extensive appendices.

**Questions:**

The following major concerns must be addressed to improve the assessment of this work:

1. Have you tested FDPW on more complex systems than Diamond or CO, such as metallic systems or systems with defects? How does the complexity of the required normalizing flow scale with the heterogeneity of the physical system?

2. Standard PW calculations typically use optimized pseudopotentials to significantly reduce the required energy cutoff. How does the efficiency of the proposed all-electron FDPW method compare to a standard PW implementation using efficient norm-conserving or PAW pseudopotentials?

3. How sensitive are the final accuracy and convergence speed to the hyperparameters of the prescribed density initialization and the elastic regularization? Was significant tuning required for the examples shown?

4. In the finite system experiments (Table 3), the PW results seem very poorly converged, and the FDPW energies also differ significantly from the reference. Could you clarify these discrepancies and the convergence behavior of both PW and FDPW for this system?

---

> ### Author Response · Authors · 2025-11-15
>
> Thank you for your thorough and insightful review. We will address each of your concerns here:
>
> Weakness:
>
> > Insufficient Comparisons to Standard Practices
>
> We acknowledge your concern. Please see common reply (R2) and (R3).
>
> > Dependence on Heuristic Initialization
>
> We acknowledge that the prescribed density is chosen using heuristics. However, this is a common practice for creating adaptive basis in the literature [Gygi, François Ab Initiomolecular Dynamics in Adaptive Coordinates, Physical Review B 51 (1995), Zumbach, Gil and Modine, N.A. and Kaxiras, Efthimios Adaptive Coordinate, Real-Space Electronic Structure Calculations on Parallel Computers, Solid State Communications 99 (1996)]. Furthermore, we would further argue that the amount of heuristics used by FDPW is much less to the PW-PP or PAW methods. The core mathematical requirement for PAW/USPP projectors is that they reproduce all-electron scattering properties (or logarithmic derivatives) up to a given angular momentum at some reference energies. This constraint underdetermines the projectors, so any practical implementation (e.g. atompaw, pslibrary, ONCVPSP, Vanderbilt's code) make heuristic design decisions, such as choice of cutoff radii for each angular momentum channel, choice of reference energies for partial waves, choice of smoothness criterion or optimization target (e.g., minimizing kinetic energy, curvature, or residual norm), number of projectors per angular momentum channel, whether to enforce norm-conservation exactly or approximately, etc. These parameters are tuned empirically for good transferability across systems, and the tuning procedure is not fully algorithmic or unique. In fact, PAW/USPP datasets from different libraries (e.g., VASP, GBRV, ONCV) differ even for the same element. As mentioned above, the basis generation procedure of FDPW is algorithmically transparent and much simpler than these methods.
>
> > However, this coordinate transformation introduces significant complexity. The Laplacian becomes the Laplace-Beltrami operator, requiring the evaluation of complex geometric quantities (Jacobian, metric tensor, connections) derived from the flow. Crucially, the Hartree potential can no longer be solved directly in Fourier space and requires an iterative Preconditioned Conjugate Gradient (PCG) solver.... Unanalyzed and Potentially Prohibitive Computational Overhead
>
> We would like to state again that one of the main merits of our methods is that, despite the use of a neural network, our method incurs *minimal overhead* compared to vanilla-PW. The key experimental result is presented in Table 1: from the speed (it/s) column, we can see that FDPW has a comparable speed to PW when using the same grid size. For example, when N=64, PW speed is 44.48 it/s and FDPW is 32.78 it/s. We attribute this minimal overhead to the following facts:
> 1. The Laplace-Beltrami operator (Eq. 18) can be evaluated using only one inverse FFT, one three-vector-valued inverse FFT, and a few point-wise multiplications (see section 4.6).
> 2. Although the Hartree energy no longer has a closed-form formula, the PCG solve can be amortized to the ground state search. With each update step, we only run 1 PCG step, and each PCG solve is warm-started from the (maybe partially converged) solution of the previous step. We show that amortization works well in Figure 5, where we compare the PCG convergence between running 10 PCG iters per DFT step, versus only running 1 PCG iter per DFT step.
> 3. As explained in (R6), the geometric quantity (distorted grid $f(\xi)$, contracted connection $A(\xi)$, inverse metric $g(\xi)$, real space external potential $V_{ext}(f(\xi)))$ required for evaluating the Laplace-Beltrami operator (see Algorithm 1) can be precomputed, and one does not need to recompute them during the DFT steps. And this precompute step is also cheap.
> 4. The density fitting step to the prescribed density is relatively fast and is independent of the ground state search. For the diamond case, it takes 72s on A100. This time can be much shorter when using a pretrained flow as explained in (R6).
>
> We would also like to reiterate that, due to the distortion, the number of grid points required in an actual calculation is naturally small.
>
> > Numerical Stability and Conditioning
>
> Strong distortions lead to ill-conditioned metric tensors -> this is not necessarily true. While strong shear distortion can lead to ill-conditioned metric tensors, we paid special attention to make sure that this does not happen by adding grid regularization (equation 15). From Figure 1, one can see that shear deformation is minimal, which is the common culprit for an ill-conditioned grid. Furthermore, we used a diagonal spectral preconditioner (described right below equation 23), which also improved numerical stability. Finally, we demonstrate the PCG convergence curve in Figure 5, and one can see that PCG converges very well.

---

> > ### Author Response · Authors · 2025-11-15
> >
> > > Limited Empirical Validation and Missing Features
> >
> > We acknowledge this point. Please see our common reply (R1) and (R5).
> >
> > > Accessibility
> > Please see (R4). We would like to add that, being an interdisciplinary work, it would not be possible to both explain the core concept rigorously while keeping it accessible to a general audience.
> >
> > Questions:
> >
> > > Have you tested FDPW on more complex systems than Diamond or CO, such as metallic systems or systems with defects? How does the complexity of the required normalizing flow scale with the heterogeneity of the physical system?
> >
> > As explained in (R1), this paper mainly tries to establish the FDPW methods. For metallic systems, we need to incorporate smearing schemes or variational optimization of occupation. This will require significant work and is typically done in follow-up papers rather than the first paper that tries to establish the method. That being said, we are actively working on extending the current framework to handle more realistic applications like metallic systems.
> >
> > > Standard PW calculations typically use optimized pseudopotentials to significantly reduce the required energy cutoff. How does the efficiency of the proposed all-electron FDPW method compare to a standard PW implementation using efficient norm-conserving or PAW pseudopotentials?
> >
> > Please see (R2).
> >
> > > How sensitive are the final accuracy and convergence speed to the hyperparameters of the prescribed density initialization and the elastic regularization? Was significant tuning required for the examples shown?
> >
> > Please see (R8).
> >
> > > In the finite system experiments (Table 3), the PW results seem very poorly converged, and the FDPW energies also differ significantly from the reference. Could you clarify these discrepancies and the convergence behavior of both PW and FDPW for this system?
> >
> > - The purpose of Table 3 is to show that FDPW converges faster than PW, and that the method works for finite systems as well. To get PW converged would require a very big grid, which we think is unnecessary since we have shown that FDPW converges with the same grid size.
> > - Crucially, we have found a bug in our scripts: when running FDPW on finite systems, we forget to add the nucleus potential energy $E_{nn}=22.5352$ to the total energy. We have updated Table 3 in the new revision to correct this mistake.

---

> > ### Comment · Reviewer_MiZ7 · 2025-11-26
> >
> > I thank the authors for their detailed responses and the effort invested in revising the manuscript. With the additional results I consider W1/2/3 solved. I do have some remaining questions/advice:
> >
> > 1. I strongly encourage the authors to integrate these arguments for utilizing normalizing flows over hand-crafted distortions (R6) more prominently into the main paper (e.g., Introduction or Section 4.3). This would significantly improve the accessibility for an ML audience and better highlight the specific ML contribution, as the current presentation remains dense and heavily reliant on domain expertise in differential geometry and physics (which unfortunately most ICLR readers do not possess).
> > 2. FDPW and PW+PAW are ultimately competing strategies for achieving computational efficiency in DFT. To demonstrate the practical impact of FDPW in the broader DFT landscape, comparisons against the established efficient paradigm (PW+PAW) are still necessary.
> > 3. The most concerning part is still the amortized PCG strategy (with iteration=1). The authors rely on Figure 5 (Appendix R) to argue that this amortization works well. I must strongly disagree with this interpretation. Figure 5 shows that the Poisson residual remains very high (1e-1) throughout the optimization. This implies the Hartree potential is highly inaccurate at every step since even after convergence the residual is still high. As such case, minimizing the energy functional with such a large, persistent error in the potential is potentially unstable. While this form of inexact optimization appears stable for the simple systems shown, for more complex systems (e.g., metals or heterogeneous materials) this might not even converge. The authors should demonstrate that this strategy converges to the same ground state as a fully converged PCG solve and provide some justifications to the stability.
> >
> > Nonetheless, I recognize the significant value of the methodological foundation presented here and I am now raising my score to 4.

---

> > > ### Author Response · Authors · 2025-12-01
> > >
> > > | Grid N | ΔE (Ha)   | Speed (it/s, 10 PCG) | Mean rel. residual (1 PCG) | Mean rel. residual (10 PCG) |
> > > |--------|-----------|---------------------|-----------------------------|------------------------------|
> > > | 32     | -0.000621 | 91.90        | 0.3095                      | 0.0497                       |
> > > | 48     | -0.001076 | 54.47        | 0.2642                      | 0.0416                       |
> > > | 64     |  0.000966 | 12.03        | 0.2713                      | 0.0465                       |
> > >
> > > We thank the reviewer for the suggestion 3 regarding the amortized PCG.
> > > We empirically validate the amortization strategy for the Hartree solve with the Diamond geometry by comparing runs with 1 PCG iteration per step against runs with 10 PCG iterations per step. As we can see in the above Table, 1 PCG runs have worse mean relative residual, but the total energy differences between 1 PCG and 10 PCG runs are <1e-3 Ha across all grid sizes. With this geometry, the Hartree potential does not need to be fully converged at the gradient descent iteration. Furthermore, we would like to add that we always run a full PCG run with an absolute tolerance of 1e-9 after the gradient descent has converged, which should greatly alleviate the self-consistency error. We acknowledge that for more heterogeneous or strongly correlated systems, a more tightly converged V_H may be necessary, and in those cases, increasing the number of PCG iterations is straightforward. Even then, the extra cost is linear in the number of PCG steps, whereas the basis-size reduction provided by FDPW lowers the number of grid points (and hence the cost of each PCG iteration) cubically. To ascertain the exact number of PCG iterations required for good convergence in difficult systems, we would need to put in more work as explained in (R1), so this remains out of the scope of this paper.

---

> > > ### Author Response · Authors · 2025-12-01
> > >
> > > We thank the reviewer for the suggestion 1 regarding the justification of using normalizing flow to create the distortion. We have added a new section (4.5) in the new revision, where we incorporated points made in the discussion (R6). We thank the reviewer again for providing valuable suggestions to our paper.

---

> ### Author Response · Authors · 2025-12-01
> **Official Comment by Authors**
>
> We thank the reviewer for the suggestion 2 regarding the comparison to PAW. To address this concern, we profiled GPAW (a well-established PW+PAW implementation) using direct optimization for a diamond crystal across varying plane-wave cutoffs. We added a figure in the appendix. Our profiling reveals two distinct computational regimes:
>
> **High-cutoff regime (theoretical):** When the plane-wave cutoff is large, the XC energy evaluation over the 3D grid via FFT dominates computational cost
> and scales as $O(N \log N)$, consistent with all-electron plane-wave methods. In this regime, a direct scaling comparison with FDPW would be appropriate.
>
> **Practical PAW regime:** However, PAW's practical advantage lies precisely in enabling *small* plane-wave cutoffs even for heavy elements—this is the
>   method's core design principle. In this regime, our profiling of diamond (a light element) already shows that the atomic XC correction cost becomes
>   comparable to the 3D grid evaluation. For heavier elements with larger augmentation spheres and higher angular momentum projectors, this atomic correction
>    is expected to become even more significant. This correction
> scales as $O(N_{\text{atom}} \times N_{\text{sphere}} \times N_{\text{radial}} \times L)$, where $N_{\text{sphere}}$ and $N_{\text{radial}}$ are the
> spherical and radial grid sizes for atomic augmentation, and $L = (2l+1)^2$ depends on the highest angular momentum $l$ of the projectors. Crucially, this
>   cost is *independent* of the plane-wave grid size.
>
> Since FDPW does not require these atomic augmentation calculations, the computational bottlenecks of the two methods are fundamentally different.
> Comparing their scalings with respect to system size or grid density would therefore not provide meaningful insight into their relative practical
> efficiency. Instead, we believe application-specific benchmarks on realistic systems would be more informative, which we plan to address in future work

---

### Official Review · Reviewer_bdxq · 2025-10-31

**Soundness:** 3
**Presentation:** 2
**Contribution:** 3
**Rating:** 6
**Confidence:** 3

**Summary:**

This paper introduces Flow‑Distorted Plane Waves (FDPW), an adaptive Galerkin basis for Kohn–Sham Density Functional Theory (DFT). The FDPW basis is constructed by applying a periodic normalizing flow on the 3‑torus to standard plane-wave coordinates. A key innovation is a modified Bloch factor, $𝑒^{ikf^{-1}(r)}$, which is designed to preserve 𝑘-point orthogonality. A special care is taken to efficiently treat different terms of arising equations.

**Strengths:**

- The paper proposes a principled way to add adaptivity to plane‑wave methods while retaining their simple algebra and FFT‑friendly structure.
- Modifying the Bloch phase to work with the distortion is neat and keeps the usual \(k\)-space structure intact.
- Using a compact normalizing‑flow map to enhance a classical solver is a sensible, lightweight use of ML that fits existing workflows.
- Preliminary experiments suggest the method maintains spectral‑like accuracy in practice while noticeably reducing parameters and resource use.

**Weaknesses:**

I see this paper as a first step in an interesting direction, but there remains substantial room for improvement. At the current stage, there is little theoretical or practical analysis of the approach’s limitations. In particular, there is no systematic study of accuracy versus cutoff or an examination of NUFFT and curvilinear discretization errors, which would help establish numerical reliability. The paper also lacks comparison with other adaptive bases, such as finite elements or wavelets, making it difficult to judge relative advantages. Despite these gaps, whether or not it becomes a powerful long-term method, in my opinion, it opens a valuable direction worth exploring, and I am leaning towards acceptance.

**Questions:**

1) Is there anything you pay for using $f^{-1}(r)$ or are there no noticiable drawbacks?
2) How does FDPW handle GGA gradients in practice? Any challenges with hybrid functionals or nonlocal pseudopotentials?
3) How sensitive are performance and accuracy to the prescribed‑density parameters $a_l,b_l,c$?

---

> ### Author Response · Authors · 2025-11-15
>
> Thank you for your insightful and encouraging review. Here we address the weakness and questions you raised.
>
> Weaknesses:
>
> > there is no systematic study of accuracy versus cutoff
>
> Table 1 provides a convergence analysis on band structure, which is essentially the accuracy versus cutoff analysis you requested. We can see much faster convergence of FDPW compared to PW
>
> > an examination of NUFFT and curvilinear discretization errors
>
> For NUFFT, please see (R7). We are not entirely sure what you mean by curvilinear discretization errors. Could you clarify this? If you are referring to the error caused by basis set incompleteness, this is already covered in Table 1.
>
> > The paper also lacks comparison with other adaptive bases, such as finite elements or wavelets
>
> Please see (R3).
>
> Questions:
>
> > Is there anything you pay for using $f^{-1}(r)$ or are there no noticiable drawbacks?
>
> Currently, the main drawback is that the density is fitted to a specific geometry, which means that if we do geometry optimization or molecular dynamics, we will need to perform the density fitting process many times. However, as explained in (R6), we are planning to counter this drawback by pretraining the flow model.
>
> > How does FDPW handle GGA gradients in practice? Any challenges with hybrid functionals or nonlocal pseudopotentials?
>
> For Hybrids and nonlocal PP, see (R5).
>
> For GGA XC functional, per particle XC energy density $\varepsilon_{\text{xc}}$ additionally depends on $|\nabla \rho|^2$, i.e. $E=\int \varepsilon_{\text{xc}}(\rho, |\nabla \rho|^2) \rho$. The functional derivative $v_{\text{xc}}$ is given by the usual Euler Lagrangian rule $v_{\text{xc}}=\frac{\delta E_{\text{xc}}}{\delta \rho} = \frac{\partial E_{\text{xc}}}{\partial\rho}  - 2\nabla \cdot  \left(\frac{\partial E_{\text{xc}} }{\partial |\nabla \rho|^{2}} \nabla \rho\right)$. The partial differentiation of  $\varepsilon_{\text{xc}}(\rho, |\nabla \rho|^2)$ to its two arguments can be handled by AD, and the pullback of the divergence and gradient operator can be derived similarly to the Laplace-Beltrami operator, which we already covered in the paper. In principle, there is no difficulty in using GGA with FDPW, but we did not include this since there is enough content for this paper already.
>
> > How sensitive are performance and accuracy to the prescribed‑density parameters
>
> Please see (R8).

---

### Author Response · Authors · 2025-11-15
**Common replies to all reviewers**

We sincerely thank all reviewers for their constructive comments and valuable suggestions that have improved the quality of our submission. Here, we address some important questions raised by reviewers from the theoretical standpoint. We will also address questions that require further experiments later on, and we will include additional results in the revised manuscripts.

(R1) Limited experimental scope:

We acknowledge this point. We would like to stress that the position of this paper is to propose the FDPW method, which can be further developed through subsequent papers. Adding support for all realistic scenarios requires substantial work, and all existing electronic structure methods start from the simple cases and gradually build up. We would like to emphasize again that our method is the first neural network ansatz that works in ab initio DFT calculation, and works for periodic systems.

(R2) Didn't compare to USPP/PAW:

We would like to emphasize the positioning of our method: our method is an *adaptive basis* powered by normalizing flow, that can be used in all-electron periodic DFT calculations with *minimal overhead*. Our method is orthogonal to PW-pseudopotentials (PW-PP) and PAW, since these methods modify the external potential, whereas our method modifies the basis. In fact, FDPW can be used in conjunction with PP/PAW:

- Using local pseudopotentials is straightforward. We already use ANC, which is a local all-electron pseudopotential
- In previous literature, like [2], a similarly constructed adaptive PW is used together with a non-local pseudopotential
- PAW methods work by decomposing the wavefunction into a smooth pseudo-wavefunction represented by plane waves and rapidly oscillating atomic-like corrections within augmentation spheres centered on each atom. It is entirely possible to replace the plane wave basis for the smooth pseudo-wavefunction with FDPW. Although formally compatible, we would like to note that combining FDPW and PAW is unnecessary since the main advantage of PAW over pseudopotentials is that it can recover all-electron properties; FDPW can already achieve this directly.

We would also like to clarify what benefits FDPW brings compared to PP/PAW:
- PW-PP methods reduce the cutoff by not modeling core electrons. This makes the applicability of PP methods limited, as it cannot model physicals property that relies heavily on core states like the NMR shielding. FDPW is all-electron: we model the core electron, and cutoff reduction comes from the distortion of physical space.
- While PAW is all-electron, it parameterizes the pseudo waves with off-line generated projectors for each atom type. The generations are completely separated from the DFT calculation, which may cause trouble if one uses HSE06 functional for DFT calculation whereas only PBE-based PAW setups are available. In contrast, the basis generation process in our method is transparent, which is a simple density fit as we described in section 4.4 and does not depend on the choice of the XC functional, and the basis is adapted to the whole system instead of atom-wise. Furthermore, the adaptation is created via a neural network, which enables subsequent works in creating data-driven methods for basis generation or other downstream tasks, which is not possible if one uses PAW.

(R3) Didn't compare to other adaptive bases:

Real-space Finite difference (FD):
1. The Laplace-Beltrami operator becomes non-SPD with naive discretization. The accuracy deteriorates quickly for nonsmooth metrics, which limits the degree of adaptivity possible.
2. The energy is no longer variational, which means that the error can carry an arbitrary sign, i.e., one could obtain an energy below the true ground-state energy.
3. Translational invariance is lost, so application to the periodic system becomes harder.
4. FD does not benefit from the FFT acceleration. In contrast, FDPW retains k-point decoupling via the modified Bloch phase factor, and we can still use FFT for acceleration.

Finite Elements (FE) and Wavelets (WL):
1. Learning-ready adaptivity: Adaptive FE/WL improves the accuracy of a calculation via grid refinement. Adaptivity is typically driven by error estimators or hand-crafted refinement and is not naturally set up for data-driven or jointly learnable basis generation. In FDPW, adaptivity is encoded by a flow model, which enables data-driven development in the future. See more in (R6).
2. Hardware and implementation: FFTs deliver very high arithmetic intensity and mature scaling on GPUs/TPUs. FE/WL matrices are sparse but non-diagonal, which pushes the cost into iterative sparse matvecs and preconditioning. Whether FE/WL outperforms PW depends on many engineering factors; we therefore chose to hold the algebra fixed and evaluate adaptivity alone. Also, FE typically has a much higher scaling constant than FFT, and its strength is mainly in handling complex boundary/geometry, which is not needed in the cases we consider.

---

> ### Author Response · Authors · 2025-11-15
>
> Lindsey & Sharma (2024) [1]:
>
> The method proposed in [1] only works for finite systems, and the main focus of the proposed methods is for periodic systems, so a direct comparison is not possible. Even so, we would argue that FDPW already gives better and more efficient kinetic calculations from theoretical considerations alone:
> - We do not perform diagonal approximation as done in [Lindsey & Sharma (2024)]. The only error comes from the spectral cutoff
> - In [Lindsey & Sharma (2024)], the kinetic evaluation requires 4 FFT and 4 inverse FFT, while in FDPW, we only need 4 inverse FFT (1 scalar value and 1 three-vector valued), as explained in section 4.6.
>
> (R4) Presentation/Accessibility
>
> We thank the reviewers for raising this point. We will revise the writing of the paper in accordance with your suggestions to increase readability. As mentioned in the paper, we will also open-source our code later, which we hope will make it much more accessible to non-mathematically inclined readers of our paper.
>
> (R5) Features not yet covered in this submission
>
> - nonlocal pseudopotentials: This is already covered in prior work [2] with a similar setup.
> - efficient force calculations: Again, this is already covered in prior work [3].
> - hybrid functionals: essentially, we need to be able to efficiently calculate exact exchange. Both the exchange operator $\hat{K}$ and the Coulomb operator $\hat{J}$ have convolutions structure: $\hat{J}f (r)=[\int \frac{1}{|r - r'|} \rho(r') d r'] f(r), \hat{K}f(r) = \sum_{n} \rho_n(r)  [\int \frac{1}{|r - r'|} \rho _n(r')f(r')dr']$, so they can both be calculated via Poisson solves (see [1]), similar to the calculation of Hartree energy which we already discussed how to do Poisson solve via PCG in Section 4.8.
>
> We would like to note that, while the literature implemented these features for finite systems, there are no fundamental difficulties for them to be adapted to the FDPW setting and to periodic systems.
>
> (R6) Why is the flow necessary, and why don't we use hand-crafted distortion?
>
> We think that the reviewers are most likely referring to hand-crafted distortion maps like those described in [3]. Here we list some of the drawbacks of such hand-crafted distortion:
> 1. The hand-crafted distortion can only represent a very narrow class of distortion, which bars it from the more interesting downstream applications like density inversion and surface modeling, which require a distortion that cannot be described by the sum of isotropic atom-centered distortions.
> 2. The approach cannot deal with a non-cubic unit cell, which is prevalent in solid-state calculations. A non-cubic unit cell essentially performs a shear transformation, and the hand-crafted transformation as described in this paper cannot handle it
> 3. The hand-crafted map is constructed as identity + atom center isotropic radial distortion. To satisfy periodic boundary conditions, one must periodize the isotropic bumps, either by summing over lattice images (Ewald‑like sums) or by crafting periodic radial kernels, which is not trivial. Both add complexity and can introduce seams/aliasing if not done perfectly.
> 4. Being an isotropic distortion, it cannot express directional features (covalent lobes, surfaces, anisotropic bonding).
>
> Specific benefits of using normalizing flow:
> 1. Using the flow construction allows us to fit the density for arbitrary input density, which is useful for downstream applications like: (a) density inversion; (b) creating distortion for surface modeling, etc.
> 2. Much better parameter to approximation power for creating the distortion, compared to the planewave distortion used in [2], or the hand-crafted distortion used in [3] (which can only represent a very narrow class of distortion)
> 3. Quantity required for the Laplace-Beltrami (LB) kernel can be easily calculated. For the LB kernel we need the contracted connection $A=\partial_\beta \log |J|$ and the inverse metric $g$. Log Jacobian determinant $\log |J|$ is readily available and cheap to calculate for all normalizing flows since it is required for evaluating probability, and its derivatives can be obtained via AD. $g$ can be obtained by applying AD to the flow bijector, which is also straightforward.
> 4. The flow model can be pretrained on randomly sampled geometries. With such a pretrained flow, the density fitting step will just need a few fine-tuning iterations, instead of the current approach, where we start from an identity flow. We are planning to do this in a future publication, so this is beyond the current scope of the paper.

---

> ### Author Response · Authors · 2025-11-15
>
> (R7) NUFFT usage
>
> We use the nufft2 function from the jax-finufft package [https://github.com/flatironinstitute/jax-finufft] with the default parameter, and eps of 1e-6. The NUFFT method is robust, and we don't see ablation necessary. Furthermore, it is only used to precompute $V_{ext}$ on the distorted grid (see Algorithms 1 and 2), so even if we don't use NUFFT, it would not affect the speed of the ground state search. NUFFT will only matter when we start to optimize geometry, or to do molecular dynamics (MD) in our planned subsequent publication.
>
> (R8) Effects of flow and density fitting hyperparameter
>
> We will add an ablation study on the flow and the density fitting hyperparameter. In particular, we will demonstrate that the metric will become ill-conditioned without proper regularization.
>
> (R9) Precompute time
>
> Here we provide the average computation time for precomputing the Laplace-Beltrami (LB) kernel and the Vext (described in Algorithm 1&2) with various grid sizes N. All computes are done on A100, and we take the average of 10 runs. JAX JIT times are reported separately.
>
> N=36
> LB: JIT 20.5s,  mean 24.5ms
> Vext: JIT 0.66s, mean 5.56ms
>
> N=48
> LB: JIT 24.17s, mean 49.1ms
> Vext: JIT 1.47s, mean 11.4ms
>
> N=64
> LB: JIT 27.75s, mean 81.5ms
> Vext: JIT 1.84s, mean 14.6ms
>
> *References*
>
> [1] Lindsey, Michael and Sharma, Sandeep Fast and Spectrally Accurate Construction of Adaptive Diagonal Basis Sets for Electronic Structure, The Journal of Chemical Physics 161 (2024).
>
> [2] Gygi, François Electronic-Structure Calculations in Adaptive Coordinates, Physical Review B 48 (1993)
>
> [3] Gygi, François Ab Initiomolecular Dynamics in Adaptive Coordinates, Physical Review B 51 (1995)

---

### Meta-Review · Area_Chair_qbws · 2025-12-09

**Summary:**

This paper proposes distorted plane waves as a building block for learning normalizing flows. This approach represents an interesting theoretical idea, which, unfortunately, is not demonstrated in terms of its pay-off for varied, practical learning scenarios.

Due to the more negative initial assessment, which most likely would have been preserved after the discussion period, I can encourage the authors to expand the empirical validation of their approach. In practice, this is important to show that the advantages of a theoretical results are preserved in practical scenarios.

**Reviewer Concerns:**

Positive aspects of the paper are:
- the periodic normalizing flow perspective, leading to the plane wave distortions,
- the use of FFTs,
- and the first promising results.

While open issues are:
- a broader empirical validation
- scaling / computational overhead aspects
- robustness / hyperparameters
- and accessibility.

**Reviewer Scores:**

Reviewer MiZ7: had a clear reject initially, raised to a 4 post-rebuttal
Reviewer bdxq: gave a positive score with a 6
Reviewer A7Dq: gave a 4 without responses

I expect that the latter two would have kept their scores.

---

### Decision · Program_Chairs · 2026-01-26

Reject